# Epigenetic drift of H3K27me3 in aging links glycolysis to healthy longevity in *Drosophila*

Zaijun Ma[1,2†], Hui Wang[1,2†], Yuping Cai[1,2†], Han Wang[1,2†], Kongyan Niu[1],
Xiaofen Wu[1,2], Huanhuan Ma[1,2], Yun Yang[1,2], Wenhua Tong[1], Feng Liu[3],
Zhandong Liu[4,5,6], Yaoyang Zhang[1], Rui Liu[7], Zheng-Jiang Zhu[1]*, Nan Liu[1]*

[1]Interdisciplinary Research Center on Biology and Chemistry, Shanghai Institute of Organic Chemistry, Pudong, China; [2]University of Chinese Academy of Sciences, Beijing, China; [3]National Research Center for Translational Medicine, State Key Laboratory of Medical Genomics, Rui-Jin Hospital, Shanghai Jiao Tong University School of Medicine, Shanghai, China; [4]Jan and Dan Duncan Neurological Research Institute, Texas Children's Hospital, Texas, United States; [5]Department of Pediatrics, Baylor College of Medicine, Houston, United States; [6]Computational and Integrative Biomedical Research Center, Baylor College of Medicine, Houston, United States; [7]Singlera Genomics, Pudong, Shanghai, China

*For correspondence:
jiangzhu@sioc.ac.cn (Z-JZ);
liunan@sioc.ac.cn (NL)

[†]These authors contributed equally to this work

**Abstract** Epigenetic alteration has been implicated in aging. However, the mechanism by which epigenetic change impacts aging remains to be understood. H3K27me3, a highly conserved histone modification signifying transcriptional repression, is marked and maintained by Polycomb Repressive Complexes (PRCs). Here, we explore the mechanism by which age-modulated increase of H3K27me3 impacts adult lifespan. Using *Drosophila*, we reveal that aging leads to loss of fidelity in epigenetic marking and drift of H3K27me3 and consequential reduction in the expression of glycolytic genes with negative effects on energy production and redox state. We show that a reduction of H3K27me3 by PRCs-deficiency promotes glycolysis and healthy lifespan. While perturbing glycolysis diminishes the pro-lifespan benefits mediated by PRCs-deficiency, transgenic increase of glycolytic genes in wild-type animals extends longevity. Together, we propose that epigenetic drift of H3K27me3 is one of the molecular mechanisms that contribute to aging and that stimulation of glycolysis promotes metabolic health and longevity.
DOI: https://doi.org/10.7554/eLife.35368.001

## Introduction

Aging is characterized by the progressive decline in cellular and organismal functions that lead to reduction of fitness and increased risks to diseases and death. Epigenetic alterations in histone modification represent a prominent hallmark of aging (*Sen et al., 2016*). Chromatin and epigenetic complexes that modify histones regulate the accessibility of DNA to transcriptional machinery, thereby permitting direct control of gene expression (*Kouzarides, 2007*). Previous studies have established that changes in histone markings, such as acetylation on histone H4 at lysine 16 (H4K16ac) in *Saccharomyces cerevisiae* (*Dang et al., 2009*), trimethylation on histone H3 at lysine 4 (H3K4me3) in *Caenorhabditis elegans* (*Han et al., 2017*), and trimethylation on histone H3 at lysine 36 (H3K36me3) in *S. cerevisiae*, *C. elegans* and *Drosophila melanogaster* substantially impact adult lifespan (*Pu et al., 2015*; *Sen et al., 2015*). Polycomb repressive complexes (PRCs), including PRC1 (*Shao et al., 1999*) and PRC2 (*Kuzmichev et al., 2002*), are important histone-modifying enzymes (*Czermin et al., 2002*; *Müller et al., 2002*; *Saurin et al., 2001*) that are evolutionarily conserved from the worm to

*Drosophila* and mammals (*Whitcomb et al., 2007*). Tri-methylated histone H3 at lysine 27 (H3K27me3) denotes transcriptional silencing, which is produced by PRC2 (*Czermin et al., 2002*; *Müller et al., 2002*) and functionally maintained by PRC1 (*Cao et al., 2002*). Although mutation in the H3K27me3 demethylase couples an increase of H3K27me3 with longevity in *C.elegans* (*Jin et al., 2011*; *Maures et al., 2011*), reducing H3K27me3 by heterozygous mutation in the H3K27me3 methyltransferase results in lifespan extension in both *C. elegans* and *D. melanogaster* (*Ni et al., 2012*; *Siebold et al., 2010*). Hence, it remains unclear whether the life-benefit phenotype is due to manipulation of specific genes or modulatory effect of H3K27me3. Importantly, H3K27me3 levels are known to increase with age (*Liu et al., 2013*; *Sun et al., 2014*); however, the manner by which H3K27me3 epigenome changes over the course of lifespan and its consequent impact on the progression of aging have not been addressed.

Metabolic homeostasis is intimately connected with aging and lifespan regulation (*López-Otín et al., 2016*). Glucose is the dominant provider of energy for cells. Glycolysis, the first step in the breakdown of glucose to extract energy in the form of ATP for cellular metabolism, is present in all cellular organisms and critical for life. The anti-aging effect of caloric restriction correlates with a metabolic feature of limiting glucose metabolism including a reduction of glycolysis (*Feuers et al., 1989*). On the other hand, glycolysis is essentially involved in a wide-range of biological processes (*Chang et al., 2013*; *Okamoto et al., 2001*; *Volkenhoff et al., 2015*). It is unclear, however, how the beneficial outcome of restricting glucose metabolism and glycolysis as a result of caloric restriction could be reconciled with the essential role of glycolysis during normal adult lifespan. Strikingly, decreased brain glucose metabolism is present prior to the onset of clinic symptoms in individuals at risk of Alzheimer's disease (*Cunnane et al., 2011*). Since aging is the most significant factor for human neurodegenerative diseases, it is important to better understand how glycolysis is modulated with age and whether modulation of glycolytic process could impact adult lifespan.

Mounting evidence associates aging with epigenetic, transcriptional, and metabolic alterations (*López-Otín et al., 2013*), yet how these processes are integrated into the regulation of adult physiology is only beginning to be revealed. Although epigenetic effects could potentially elicit broad regulatory mechanisms, it remains to be determined whether epigenetic modulations are simply associated with aging or directly contribute to transcriptional and metabolic modulations that profoundly impact adult lifespan. Here, using *Drosophila*, we discover that adult-onset fidelity loss results in epigenetic drift of H3K27me3, which in turn impinges on the transcription and metabolism of the glycolytic pathway. Our data presented in this study are consistent with the model that epigenetic drift of H3K27me3 is one of the molecular mechanisms that contribute to at least a subset of age-associated phenotypes and limit lifespan in *Drosophila*. Moreover, we reveal that a reduction of H3K27me3 by PRC2-deficiency stimulates glycolysis and healthy lifespan. Our study proposes that, during natural aging, stimulation of glycolysis promotes metabolic health and longevity.

## Results

### Adult-onset fidelity loss results in epigenetic drift of H3K27me3

H3K27me3 is a signature histone modification, but its dynamics at the epigenome level during aging has not been studied. To characterize H3K27me3 during aging, we examined the level of H3K27me3 in muscle by western blot and revealed an age-associated increase in wild-type (WT) flies (*Figure 1—figure supplement 1A*). To quantitatively profile H3K27me3, we adapted the chromatin immunoprecipitation (ChIP) followed by high-throughput DNA sequencing with reference exogenous genome method (ChIP-Rx) (*Bonhoure et al., 2014*; *Orlando et al., 2014*) in fly muscle tissues (*Figure 1—figure supplement 1B*). We generated H3K27me3 epigenome profiles in adult flies of 3, 15, and 30d. In addition, we also sampled embryo, larvae, and pupae to examine H3K27me3 epigenome profiles during development. Surprisingly, despite the fact that H3K27me3 levels increased with age, H3K27me3 peak profiles in the aging adult remained largely unchanged (*Figure 1A*). In a sharp contrast, H3K27me3 signals during developmental stages showed dramatic gain or loss (*Figure 1B*). This evidence highlights a distinct, yet unknown mechanism of H3K27me3 modification that occurs during aging.

To define the landscape of aging epigenome, we divided distribution of H3K27me3 into peak regions (IP/input $\geq$ 2 and signal spanning $\geq$3 kb) and inter-peak regions. Analysis in 3d WT revealed

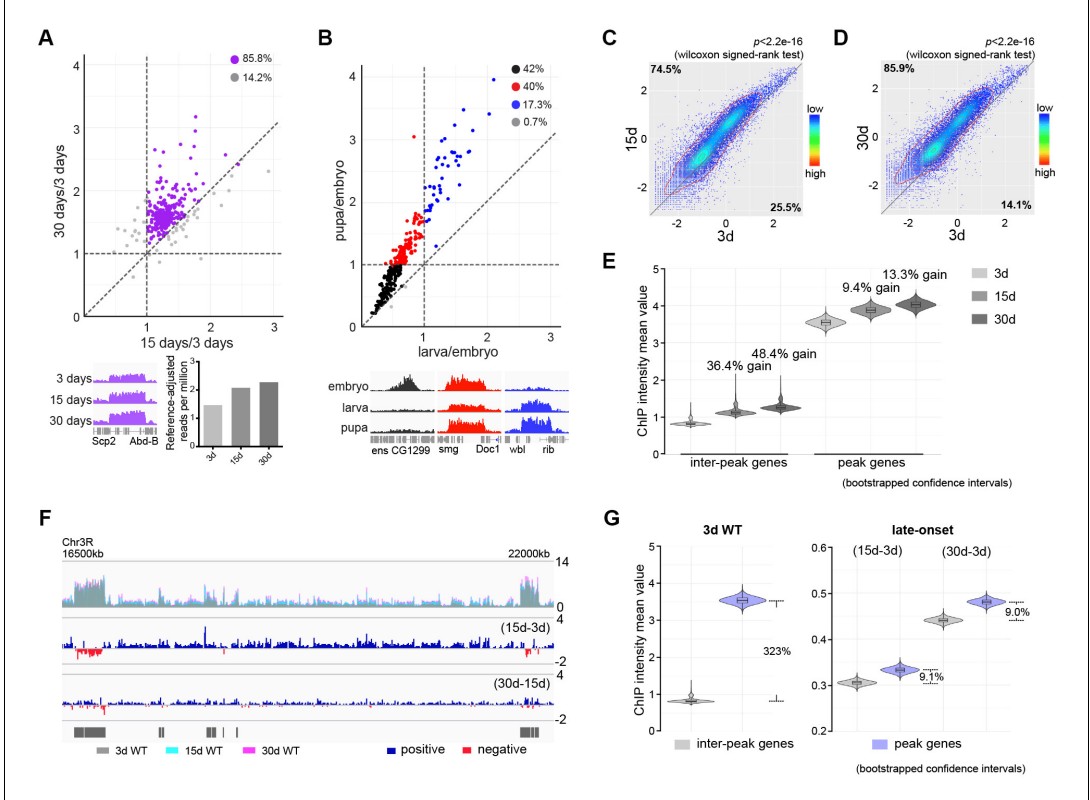

**Figure 1.** Adult-onset fidelity loss results in epigenetic drift of H3K27me3. (**A**) H3K27me3 peak profiles are comparable between young and aged animals. A scatter plot (top panel) of 331 peaks and a genome browser mini-view (bottom, left panel) illustrated that H3K27me3 had comparable peak profiles between young and aged animals. Peaks were identified from four biological replicates of 3d and 30d male flies using Homer. Data plotted are mean values of peak region at 15d (X-axis) or 30d (Y-axis) as compared to 3d. Each dot on the plot represents a peak locus. For quantitative comparison, the dm6 mapped reads were normalized to a scale factor using deeptools function bamCoverage (see Materials and methods for further details). Majority of peak regions were well maintained during aging (purple). Bar chart (bottom, right panel) showed that the number of Reference-adjusted Reads Per Million (RRPM) using ChIP-Rx datasets, exhibiting a progressive increase of H3K27me3 level with age. ChIP-seq was from muscle tissues of 3-, 15-, and 30d-old male flies. Genotype: 5905. (**B**) H3K27me3 modification undergoes dramatic gain or loss during development. A scatter plot (top panel) of 300 peaks showed dynamic changes of H3K27me3 signals during embryo, larvae, and pupae. Peaks were identified from two biological replicates of three developmental stages. Data plotted are mean values of peak region at larva (X-axis) or pupa (Y-axis) as compared to embryo. Each dot on the plot represents a peak locus. A scale factor normalized dm6 mapped reads were used to generate the mean value of each peak. Method same as (**A**). As illustrated by the genome browser mini-view for select genes (bottom panel), H3K27me3 modification was either embryo-specific (black) or selectively decreased in larvae (red), or progressively increased from embryo, larvae to pupae (blue). ChIP-seq was from whole embryo, larvae, and pupae. Genotype as in (**A**). (**C**) and (**D**) H3K27me3 modification increases with age. Scatter plot showed H3K27me3 levels for protein-coding genes (transcriptional start site to transcriptional termination sites as annotated in dm6) of in 3d as compared to 15d (**C**) and 30d (**D**). (wilcoxon signed-rank test, p<2.2e-16). The dm6 mapped reads were normalized to a scale factor to compare the relative H3K27me3 level quantitatively. Each dot on the plot represents a single gene locus. X- and Y-axis represented log2 mean value of gene's reference-adjusted reads. Contour lines indicated that H3K27me3 signals were higher in aged flies compared to 3d-old flies. ChIP-seq and genotype as in (**A**). (**E**) Inter-peak genes receive relatively more H3K27me3 modification during aging. We used bootstrapping to generate the mean reference-adjusted ChIP intensity of peak genes or inter-peak genes (10,000 draws with replacement of n = 500). Violin plots represent the bootstrapped mean H3K27me3 level of inter-peak genes (left) and peak genes (right) at 3d, 15d, and 30d. In aging flies, inter-peak genes gained relatively more signal (15d: 36.4%, 30d: 48.4%) than peak genes (15d: 9.4%, 30d: 13.3%). Net gain of H3K27me3 signals during aging was calculated by a subtraction of the mean signal intensity between aged and 3d. Bootstrapped 95% confidence intervals: 3d inter-peak genes: [0.758, 1.017]; 15d inter-peak genes: [1.023, 1.401]; 30d inter-peak genes: [1.149, 1.562]; 3d peak genes: [3.349, 3.739]; 15d peak genes: [3.681, 4.075]; 30d peak genes: [3.817, 4.230]. ChIP-seq and genotype as in (**A**). (**F**) H3K27me3 modification during aging has reduced selectivity. Genome browser view of a 5.5 Mb region in the chromosome 3R was shown. H3K27me3 occupancy overlaid between 3d (grey), 15d (cyan), and 30d (pink) (top panel). H3K27me3 modification during aging was shown by deducting 3d signals from those at 15d (middle panel) and by deducting 15d signals from those at 30d of age (bottom panel). The increased modifications had no direct correlation with pre-existing peaks. Black boxes and lines represented peak regions, corresponding to their chromosomal locations. ChIP-seq and genotype as in (**A**). (**G**) Violin plots indicate that signals are highly preferential at the peak regions at 3d (left, about 323% more signal compared to inter-peak), but for signals gained during aging, selectivity is dramatically reduced (right, only about 9% more signal acquired by peak than inter-peak). Method same as (**E**). The proportion of more signals in peak genes was computed by a subtraction of the mean signal intensity between peak genes

*Figure 1 continued on next page*

*Figure 1 continued*

and inter-peak genes. Bootstrapped 95% confidence intervals: 3d inter-peak genes: [0.758, 1.017]; 3d peak genes: [3.349, 3.739]; late-onset (15d-3d) inter-peak genes: [0.295, 0.315]; (15d-3d) peak genes: [0.320, 0.345]; (30d-3d) inter-peak genes: [0.430, 0.452]; (30d-3d) peak genes: [0.468, 0.493]. ChIP-seq and genotype as in (A).

DOI: https://doi.org/10.7554/eLife.35368.002

The following figure supplement is available for figure 1:

**Figure supplement 1.** Global occupancy of H3K27me3 modification is preserved with age.

DOI: https://doi.org/10.7554/eLife.35368.003

222 peak regions reproducibly identified from a total of four replicates, which covered approximately 9.73 Mbp of the genome (*Figure 1—figure supplement 1C*, *Supplementary file 1*). Analysis of global occupancy of H3K27me3 between young and aged WT exhibited comparable patterns (pair-wise Pearson correlation coefficients $\geq$ 0.70) (*Figure 1—figure supplement 1D*), demonstrating that the genome-wide distributions of H3K27me3, particularly the peak regions, are maintained in aged WT.

We next analyzed signal intensity of individual genes using reference-adjusted read counts. Scatter plots demonstrated that H3K27me3 signals in WT clearly had an adult-onset, progressive increase (*Figure 1C and D*). To further characterize H3K27me3 dynamics, we sorted individual genes into peak genes (1141 genes) according to the peak regions and genes therein covered (see *Supplementary file 1* for peak regions and included genes) or inter-peak genes (16370 genes). In aging WT flies, despite a genome-wide increase, we noted that inter-peak genes gained relatively more signals than those of peak genes (*Figure 1E*). To quantify the H3K27me3 marks gained specifically during aging, we subtracted the signals of 3d from those of older age points. Genome browser views illustrated that subtracted signals propagated along the chromosome, showing no direct correlation with pre-existing peaks (*Figure 1F*). We then extended this analysis to the entire epigenome. Modification in flies that were 3d old was predominantly biased at the peaks, with approximately 323% more signals as compared to the inter-peaks (*Figure 1G*). In contrast, the signals acquired during aging were only about 9% more at the peaks than those at the inter-peaks (*Figure 1G*), suggesting that there was a dramatic shift in the pattern of H3K27me3 modification in aging. Altogether, analysis of H3K27me3 aging dynamics implicates a strong selection on peak regions compared to inter-peak regions when animals are young; however, adult-onset modification of H3K27me3 becomes less biased at the peak regions.

## A CRISPR/Cas9 deficiency screen identifies the role of PRCs in aging

Given that above study indicated a new feature of H3K27me3 dynamics at the epigenome level during adult lifespan, we further explored the roles of epigenetic pathways in the aging process via an unbiased deficiency-based genetic screen. Using CRISPR/Cas9 mutagenesis (*Ren et al., 2013*), we systematically generated site-specific deletion mutants of 24 key regulators involved in distinct epigenetic and chromatin modifications in the same homogenous genetic background (*Supplementary file 2*). Since the effects of epigenetic genes are usually dose-dependent, we analyzed adult survival using heterozygous mutants. Lifespan analysis demonstrated that whereas majority of mutants showed mild or no effect, animals deficient in PRCs, including *esc*, *E(z)*, *Pcl*, *Su(z)12* of PRC2 and *Psc* and *Su(z)2* of PRC1, lived substantially longer (*Figure 2A*, *Figure 2—figure supplement 1A*, *Supplementary file 2*). Roles of select PRC2 factors have been previously implicated in *Drosophila* lifespan regulation (*Siebold et al., 2010*), although the underlying mechanism remains to be addressed. In addition, it was still possible that life-extension might be due to manipulations of individual genes, but not linked to the epigenetic change of H3K27me3. We further examined pair-wise combinations of PRC2 double mutants in trans-heterozygosity. Remarkably, double mutants demonstrated more striking effects on H3K27me3-reduction and life-extension (*Figure 2B*, *Figure 2—figure supplement 1B*). Identification of multiple components of PRC2 and PRC1 in the screen is interesting, suggesting the involvement of the epigenetic changes regulated by these two complexes in animal lifespan.

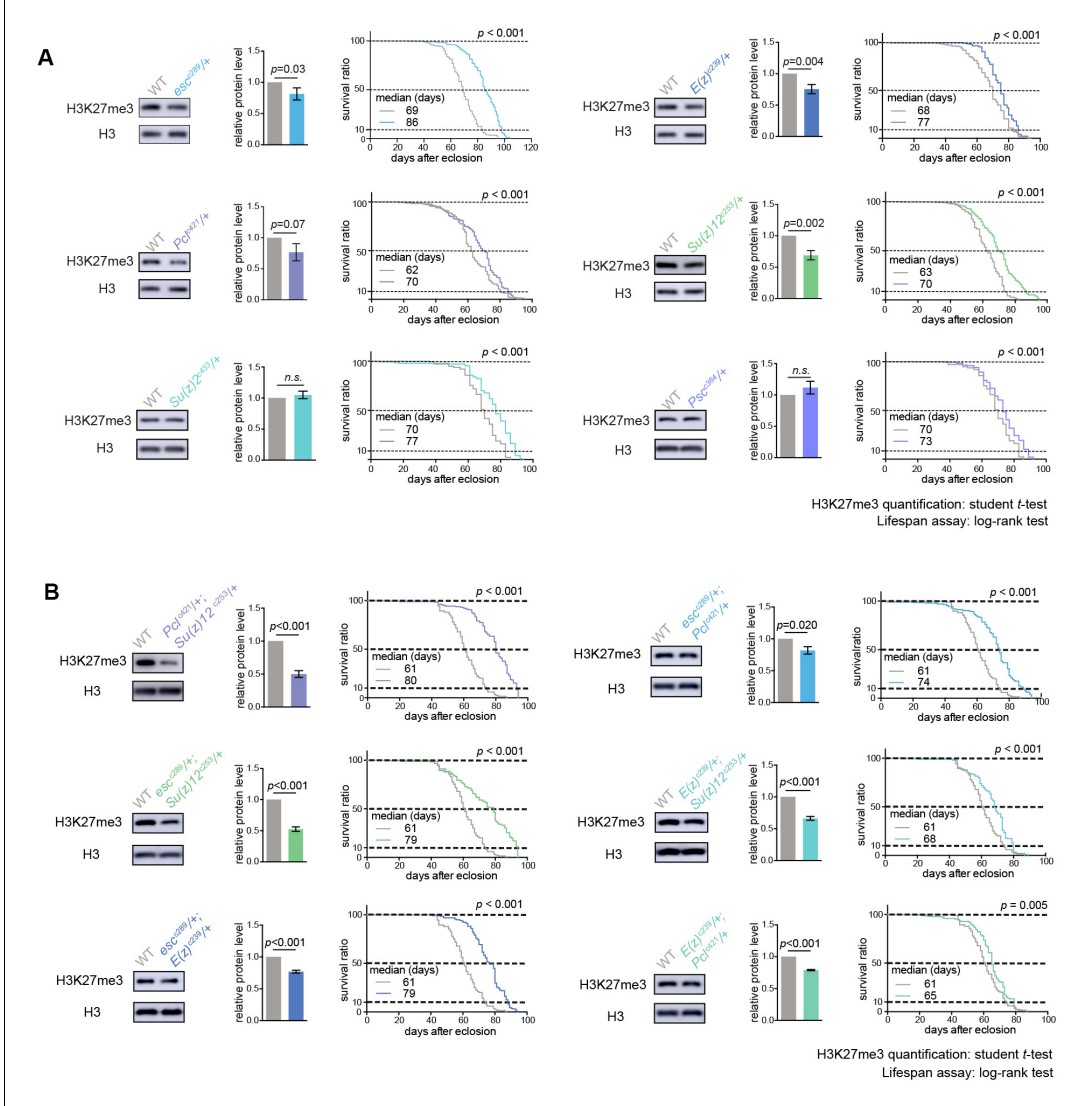

**Figure 2.** PRCs-deficient animals have extended lifespan. (**A**) PRCs-deficient animals have extended lifespan. H3K27me3 western blot (left), H3K27me3 quantification (middle), and lifespan curve (right) for PRC single mutants. PRC2 heterozygous mutants of indicated genotype reduce H3K27me3 levels and extend lifespan; PRC1 heterozygous mutants of indicated genotype extend lifespan without changing H3K27me3 levels. To name new mutant, a superscript amended to the gene contained a letter c denoting CRISPR/Cas9 method followed by the size of genomic deletion. All mutants have been backcrossed with WT for five times to ensure a uniform genetic background. See also *Figure 2—figure supplement 1A*. Western blot was from head tissues of 3d-old male flies. (for H3K27me3 quantification: mean ±SD of three biological repeats; student *t*-test; for lifespan assay: 25°C; n ≥ 200 per genotype for curve; log-rank test). (**B**) Pair-wise combination of PRC2 trans-heterozygous double mutants of indicated genotype results in stronger effects in H3K27me3-reduction and life-extension. See also *Figure 2—figure supplement 1B*. Western blot was from head tissues of 3d-old male flies. (for H3K27me3 quantification: mean ±SD of three biological repeats; student *t*-test; for lifespan assay: 25°C; n ≥ 200 per genotype for curve; log-rank test).

DOI: https://doi.org/10.7554/eLife.35368.004

The following figure supplement is available for figure 2:

**Figure supplement 1.** PRCs-deficient animals have extended lifespan.

DOI: https://doi.org/10.7554/eLife.35368.005

## Long-lived PRC2 mutants diminish the epigenetic drift of H3K27me3 during aging

We then investigated whether the age-delaying effect of PRCs-deficiency might be due to its ability to modulate H3K27me3 dynamics with age. Using muscle tissues, we examined the modification of

H3K27me3 in young (3d) and aged (30d) mutants with their age-matched WT controls. For this and subsequent experiments, we studied the PRC2 deficiency using $Pcl^{c421}$ $Su(z)12^{c253}$ trans-heterozygote double mutants, which gave rise to the strongest effect in H3K27me3-reduction and life-extension. Of note, western blot analysis showed that only H3K27me2/3 were selectively reduced in PRC2 mutants, while other histone markings remained unchanged (*Figure 3—figure supplement 1A*). Interestingly, global occupancy of H3K27me3 between WT and age-matched PRC2 mutants was preserved, as demonstrated by the comparable patterns of peak profiles and genes known to be H3K27me3-modified (pair-wise Pearson correlation coefficients ≥ 0.77) (*Figure 3A*, *Figure 3—figure supplement 1B and C*), although we cannot rule out the possibility that certain H3K27me3 targets might be altered. H3K27me3 signals in WT were generally higher than those in mutants (*Figure 3B*, see *Figure 3—figure supplement 1D* for $esc^{c289}$ $E(z)^{c239}$). In PRC2 mutants at 3d where H3K27me3 bulk levels were substantially decreased, interestingly, the signals became more likely to coalesce into the peak regions (*Figure 3C*, see *Figure 3—figure supplement 1E* for $esc^{c289}$ $E(z)^{c239}$). Given limited H3K27me3 repertoire in the mutants, this shift might be critical for the establishment of the peaks. In aging PRC2 mutants, adult-onset modification was less biased or even decreased at the

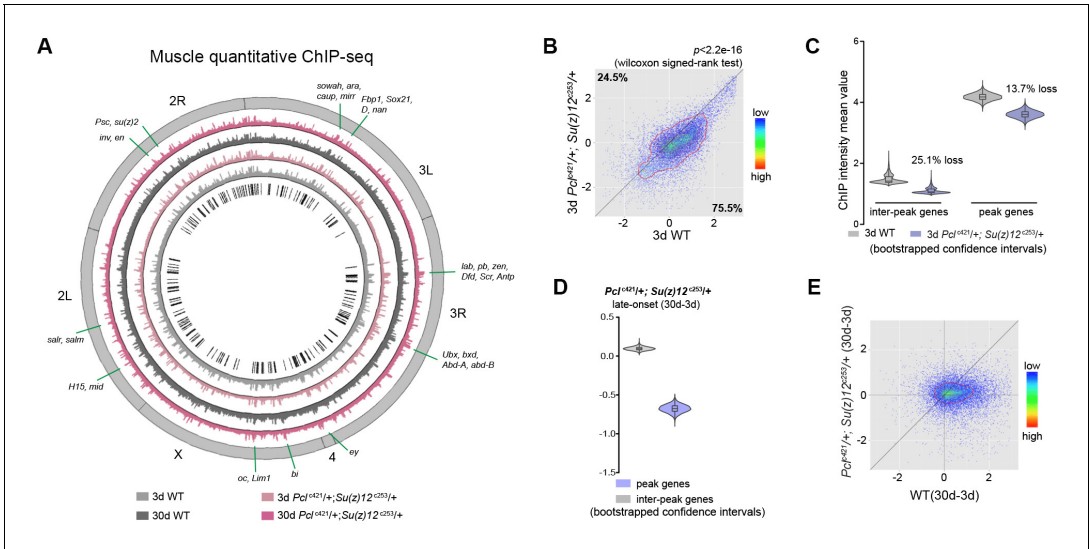

**Figure 3.** Long-lived PRC2 mutants diminish the epigenetic drift of H3K27me3 during aging. (A) Circos plot of the H3K27me3 epigenome illustrates peak profiles that are highly preserved with age and in PRC2 mutants. Black boxes and lines (innermost circle) represented common peak regions, corresponding to their chromosomal locations. Chromosome ideogram was in grey (outermost ring). PRC2 target genes previously found in cells and during development were shown next to their epigenomic loci. ChIP-seq was from muscle tissues of 3d- and 30d-old male flies. Genotypes: WT: 5905 and $Pcl^{c421}/+$; $Su(z)12^{c253}/+$. (B) H3K27me3 modification decreases in PRC2 mutants. Scatter plot showed H3K27me3 levels for gene bodies of all protein-coding genes in $Pcl^{c421}$; $Su(z)12^{c253}$ as compared to WT. (wilcoxon signed-rank test, p<2.2e-16). Each dot on the plot represents a single gene locus. X- and Y-axis represented log2 mean value of genes' reference-adjusted reads. Contour lines indicated that H3K27me3 signals were generally higher in WT compared to mutants. ChIP-seq was from muscle tissues of 3d-old male flies. Genotypes as in (A). (C) Inter-peak genes receive relatively less H3K27me3 modification in PRC2 mutants. In PRC2 mutant, inter-peak genes received relatively less signal (25.1%) than those of peak genes (13.7%). Reduction of signals was calculated by a subtraction of the mean signal intensity between WT and PRC2 mutants. Bootstrapped 95% confidence intervals: 3d inter-peak genes: [1.317, 1.861]; 3d $Pcl^{c421}$; $Su(z)12^{c253}$ inter-peak genes: [0.944, 1.361]; 3d peak genes: [3.946, 4.414]; 3d $Pcl^{421}$; $Su(z)12^{253}$ peak genes: [3.364, 3.861]. ChIP-seq and genotypes as in (A). (D) Modification of H3K27me3 during aging has reduced selectivity in PRC2 mutants. Bootstrapped 95% confidence intervals: $Pcl^{421}$; $Su(z)12^{253}$ late-onset inter-peak genes: [0.053, 0.147]; $Pcl^{421}$; $Su(z)12^{253}$ late-onset peak genes: [−0.778,−0.578]. ChIP-seq was from muscle tissues of 3d and 30d old male flies. Genotypes: $Pcl^{c421}/+$; $Su(z)12^{c253}/+$. (E) Age-associated drifting of H3K27me3 is dampened in PRC2 mutants. Scatter plot showed gained H3K27me3 signal for all protein-coding genes in aged $Pcl^{c421}$; $Su(z)12^{c253}$ and WT flies compared to young flies. Each dot on the plot represents a single gene locus. X- and Y-axis represented signal intensity transformed by Log2. Contour lines indicated that the majority of the gene signals displayed higher levels—thus more rapid changes—in aged WT compared to mutants. ChIP-seq and genotypes as in (A).

DOI: https://doi.org/10.7554/eLife.35368.006

The following figure supplement is available for figure 3:

**Figure supplement 1.** Long-lived PRC2 mutants diminish the epigenetic drift of H3K27me3 during aging.

DOI: https://doi.org/10.7554/eLife.35368.007

peaks (*Figure 3D*, see *Figure 3—figure supplement 1F* for *esc*[c289] *E(z)*[c239]). More strikingly, age-associated shift in H3K27me3 modification that occurred in WT was now dampened in mutants (*Figure 3E*, see *Figure 3—figure supplement 1G* for *esc*[c289] *E(z)*[c239]). This evidence potentially indicates that PRC2-mutation mitigates the drifting of H3K27me3 that occurs during natural aging.

To extend this analysis beyond muscle tissues, we further profiled H3K27me3 using dissected heads of *Pcl*[c421] *Su(z)12*[c253] mutants and their age-matched WT controls. Similarly, our data shows that H3K27me3 in heads has age-associated increase and drift during normal aging in WT animals, and that this shift can be partially diminished by PRC2-mutation (*Figure 4*, *Figure 4—figure supplement 1*).

## Transcriptomics links H3K27me3 dynamics to the regulation of glycolytic genes

Since biological activities of histone modification often link chromatin accessibility to gene expression (*Kouzarides, 2007*), we speculated that alterations in the levels of H3K27me3, a repressive epigenetic mark in transcription, could impact the transcriptome. We asked whether transcriptional change of particular genes might account for PRC2-dependent life-extension. Using muscle tissues, we generated RNA-seq datasets for polyA-selected mRNAs. Analysis of individual PRC2 mutants resulted in several hundreds of upregulated genes, most of which were those at the inter-peak regions (*Figure 5—figure supplement 1A*). On the other hand, analysis of 63 genes with known effects on aging, including genes in insulin/IGF-1, mTOR pathways, etc., revealed no consistent changes in the expression in PRC2 mutants (*Supplementary file 3*), thus excluding them from being the major contributors in mediating PRC2-dependent longevity. Interestingly, comparative transcriptomics converged on a common set of genes (*Figure 5A*, *Supplementary file 3*). Gene ontology (GO) analysis revealed that the 'glycolytic process' and closely related pathways were highlighted for genes upregulated (*Figure 5A*), while the 'oxidation-reduction process' was enriched for genes downregulated (*Figure 5A*). Using weighted gene co-expression network analysis (WGCNA) (*Langfelder and Horvath, 2008*), we determined the significance of highly correlated changes in the expression of glycolytic genes (*Figure 5B*). We then extended RNA-seq analysis to the heads, and further narrowed down to two glycolytic genes, *Tpi* and *Pgi*, whose expressions were upregulated in both tissue types across individual long-lived PRC2 mutants (*Figure 5—figure supplement 1B and C*, *Supplementary file 4*). As noted, glucose is metabolized through sequential reactions of glycolysis and the citric acid cycle. However, we found no consistent changes in the expression of genes mediating pentose phosphate pathway (PPP), citric acid cycle, and oxidative phosphorylation (*Figure 5—figure supplement 1C*, *Supplementary file 4*), suggesting modulatory effects on the glycolytic genes.

Moreover, RNA-seq analyses of WT animals revealed that, although aging might exert a much broader effect on the changes of the transcriptome, genes downregulated with age were mainly enriched in carbohydrate metabolism and energy production (*Figure 5C*). In particular, we noted a transcriptional decline of genes related to glycolysis/gluconeogenesis in old animals (*Figure 5D*). To verify these results, we subsequently performed single-gene analysis on *Tpi* and *Pgi* with respect to their H3K27me3 states and mRNA levels using ChIP-qPCR (*Figure 5E*, *Figure 5—figure supplement 1D*) and qRT-PCR (*Figure 5F and G*), respectively. In addition to PRC2 mutants, we attested the transcriptional increase of these two glycolytic genes in *Su(z)2*[c433], a long-lived PRC1 mutant (*Figure 5H*).

Given the fact that H3K27me3 naturally increased with age, we investigated whether further increase of H3K27me3 could impact lifespan and gene expression. To address this, we examined flies deficient in histone demethylation. The *Drosophila Utx* gene has been previously implicated in catalyzing the removal of methyl groups from H3K27me3 (*Smith et al., 2008*); a null mutation, *utx*[c499] (*Supplementary file 2*), displayed a further increase of H3K27me3 compared to age-matched WT (*Figure 5E and I*, *Figure 5—figure supplement 1D*). Flies lacking *utx* were viable, but detailed characterization indicated a significantly shortened lifespan, to about 70% that of WT animals (*Figure 5J*). Concomitant with an increase in H3K27me3 modification, expression of both *Tpi* and *Pgi* was further decreased (*Figure 5K*). Finally, control studies using short-lived *miR-34* mutants (*Liu et al., 2012*) and a long-lived *piwi* deficiency that was identified from our unbiased genetic screen (*Supplementary file 2*) demonstrated that transcriptional changes of the glycolytic genes had no simple correlation with lifespan alterations (*Figure 5—figure supplement 1E and F*).

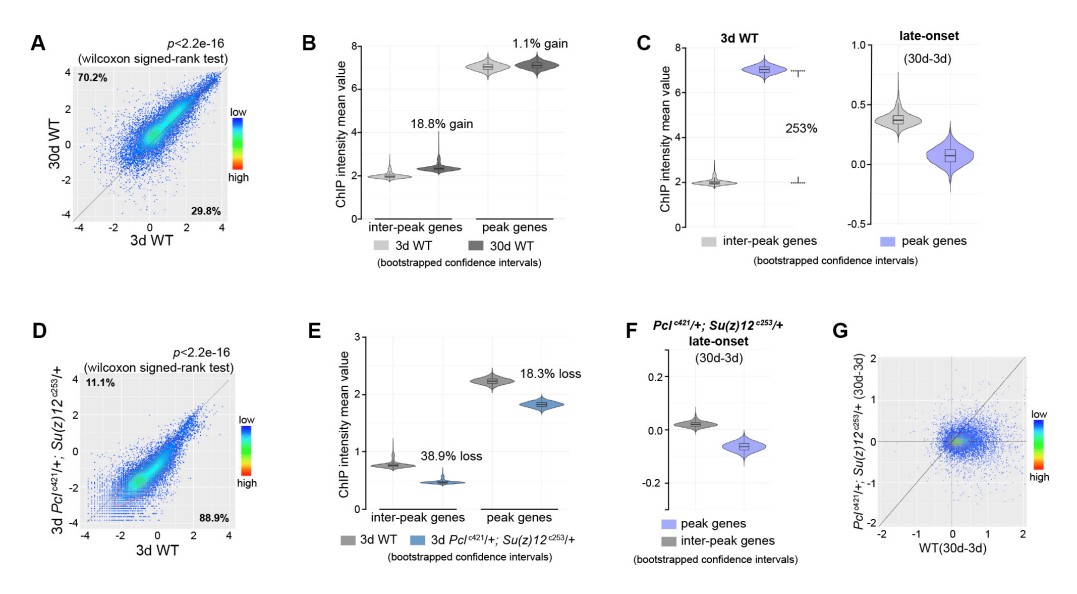

**Figure 4.** Epigenetic drifting of H3K27me3 occurs with age in head. (**A**) H3K27me3 modification increases with age in head. Scatter plot showed H3K27me3 levels for protein-coding genes (transcriptional start site to transcriptional termination sites as annotated in dm6) of in 3d as compared to 30d. (wilcoxon signed-rank test, p<2.2e-16). The dm6 mapped reads were normalized to a scale factor to quantitatively compare the relative H3K27me3 level. Each dot on the plot represents a single gene locus. X- and Y-axis represented log2 mean value of gene's reference-adjusted reads. Contour lines indicated that H3K27me3 signals were generally higher in aged compared to 3d-old flies. ChIP-seq was from head tissues of 3d- and 30d-old male flies. Genotype: 5905. (**B**) Inter-peak genes receive relatively more H3K27me3 modification during aging. We used bootstrapping to generate the mean reference-adjusted ChIP intensity of peak genes or inter-peak genes (10,000 draws with replacement of n = 500). Violin plots represent the bootstrapped mean H3K27me3 level of inter-peak genes (left) and peak genes (right) at 3d and 30d. In aging flies, inter-peak genes gained relatively more signal (30d: 18.8%) than peak genes (30d: 1.1%). Net gain of H3K27me3 signals during aging was calculated by a subtraction of the mean signal intensity between aged and 3d. Bootstrapped 95% confidence intervals: 3d inter-peak genes: [1.907, 2.027]; 30d inter-peak genes: [2.263, 2.402]; 3d peak genes: [6.912, 7.146]; 30d peak genes: [6.985, 7.217]. ChIP-seq and genotype as in (**A**). (**C**) Violin plots indicate that signals are highly preferential at the peak regions at 3d (left), but for signals gained during aging, selectivity is dramatically reduced (right). Statistics analysis as in (**B**). Bootstrapped 95% confidence intervals: 3d inter-peak genes: [1.907, 2.027]; 3d peak genes: [6.912, 7.146]; late-onset (30d-3d) inter-peak genes: [0.2756, 0.505]; (30d-3d) peak genes: [−0.136,0.248]. ChIP-seq and genotype as in (**A**). (**D**) H3K27me3 modification decreases in PRC2 mutants. Scatter plot showed H3K27me3 levels for gene bodies of all protein-coding genes in $Pcl^{c421}$; $Su(z)12^{c253}$ as compared to WT. (wilcoxon signed-rank test, p<2.2e-16). Each dot on the plot represents a single gene locus. X- and Y-axis represented log2 mean value of gene's reference-adjusted reads. Contour lines indicated that H3K27me3 signals were generally higher in WT compared to mutants. ChIP-seq was from head tissues of 3d-old male flies. Genotypes: WT: 5905 and $Pcl^{421}/+$; $Su(z)12^{c253}/+$. (**E**) Inter-peak genes receive relatively less H3K27me3 modification in PRC2 mutants. In PRC2 mutant, inter-peak genes received relatively less signal (38.9%) than those of peak genes (18.3%). Statistics analysis as in (**B**). Bootstrapped 95% confidence intervals: 30d WT inter-peak genes: [0.737, 0.783]; 30d $Pcl^{421}$; $Su(z)12^{c253}$ inter-peak genes: [0.4498, 0.4787]; 30d WT peak genes: [2.193, 2.268]; 30d $Pcl^{421}$; $Su(z)12^{253}$ peak genes: [1.787, 1.858]. ChIP-seq and genotypes as in (**D**). (**F**) Modification of H3K27me3 during aging has reduced selectivity in PRC2 mutants. Statistics analysis as in (**B**). Bootstrapped 95% confidence intervals: $Pcl^{421}$; $Su(z)12^{253}$ late-onset inter-peak genes: [0.0126, 0.026]; $Pcl^{421}$; $Su(z)12^{253}$ late-onset peak genes: [−0.076,–0.052]. ChIP-seq was from head tissues of 3d and 30d old male flies. Genotypes: $Pcl^{c421}/+$; $Su(z)12^{c253}/+$. (**G**) Age-associated drifting of H3K27me3 is dampened in PRC2 mutants. Scatter plot showed gained H3K27me3 signal for all protein-coding genes in aged $Pcl^{c421}$; $Su(z)12^{c253}$ and WT flies compared to young flies. Each dot on the plot represents a single gene locus. X- and Y-axis represented signal intensity. Contour lines indicated that the majority of the gene signals displayed higher levels—thus more rapid changes—in aged WT compared to mutants. ChIP-seq was from head tissues of 3d and 30d-old male flies. Genotypes: WT: 5905 and $Pcl^{c421}/+$; $Su(z)12^{c253}/+$.
DOI: https://doi.org/10.7554/eLife.35368.008

The following figure supplement is available for figure 4:

**Figure supplement 1.** Global occupancy of H3K27me3 modification in head tissues with age and in PRC2-deficiency.
DOI: https://doi.org/10.7554/eLife.35368.009

Altogether, these data reveal that H3K27me3 dynamics with natural aging couples with a downregulation of glycolytic genes, and this reduction is enhanced in *utx* mutants upon further increase of H3K27me3. Long-lived PRC2 mutants partially reverse the decline in the expression of glycolytic genes that are otherwise significantly decreased with age.

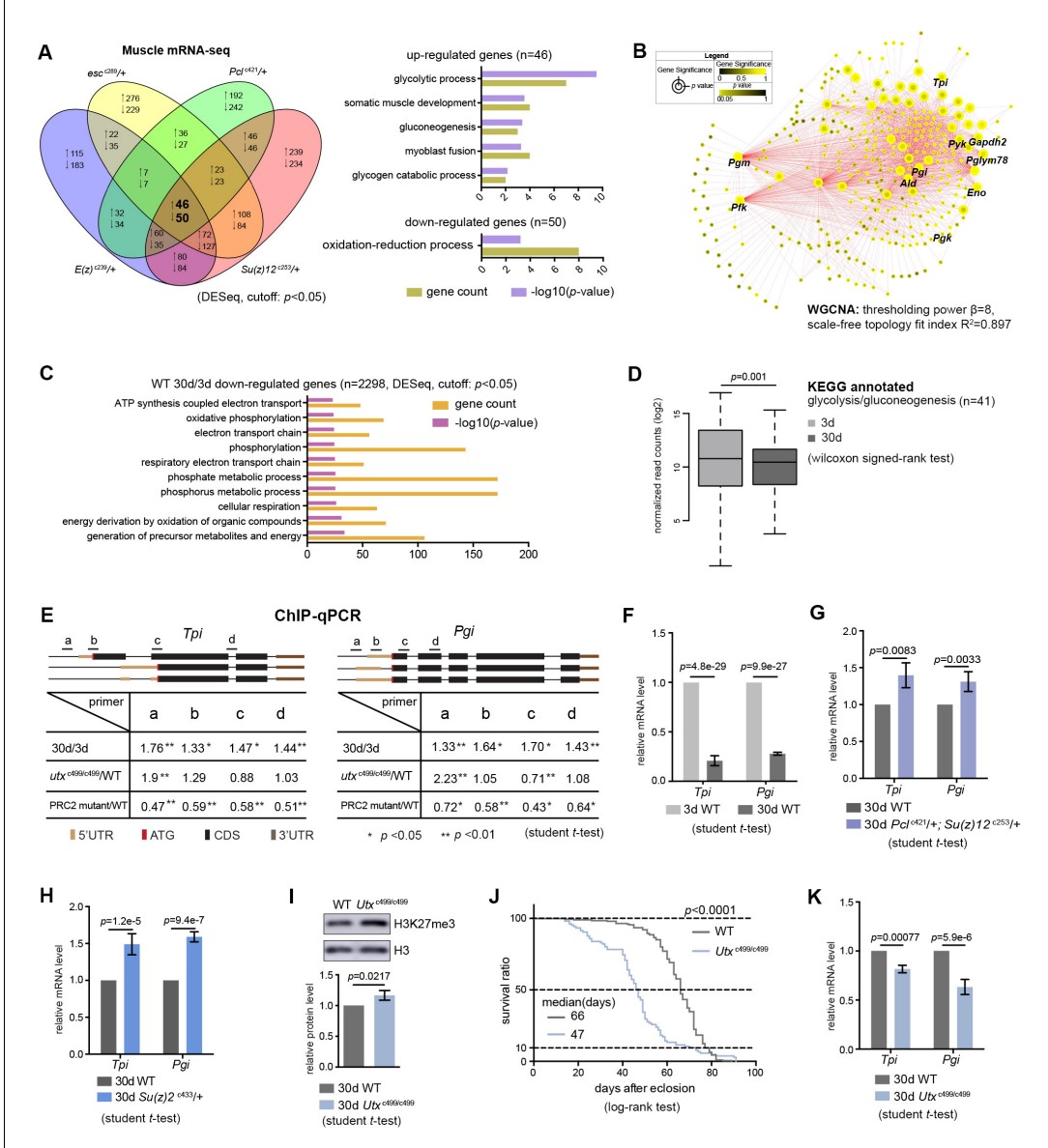

**Figure 5.** Transcriptomics links H3K27me3 dynamics to the regulation of glycolytic genes. (**A**) Venn diagram shows genes commonly changed in PRC2 single mutants with indicated genotype (left panel). Differential expression genes were computed using DESeq based on normalized count from three biological replicates (p<0.05). Gene ontology (GO) analysis (performed by David) shows glycolysis being the biological processes significantly enriched for genes upregulated in PRC2 long-lived mutants, while oxidation-reduction process is the only pathway enriched for genes down-regulated. The bar graphs represent –log10 (p value) and gene counts in each pathway. RNA-seq was from muscle tissues of 30d-old male flies. (**B**) WGCNA network analysis reveals a co-regulated change of glycolytic genes. Glycolytic genes were highlighted. WGCNA was used for finding modules of highly correlated genes across mRNA-seq samples. For selected module, Cytoscape was used for visualizing interaction network. Glycolytic genes were highlighted as node, suggesting their expression was changed coordinately. (**C**) GO analysis shows biological processes related to energy metabolism being significantly decreased with age in WT. Age-dependent mRNA expression change was computed using DESeq based on mRNA-seq data from three biological replicates. GO analysis of 2298 down-regulated genes (cutoff: p<0.05) was performed by David. The top 10 most significantly affected biological processes were shown. The bar graphs represent –log10 (p value) and gene counts in each pathway. RNA-seq was from muscle tissues of 3d- and 30d-old male flies. Genotype: 5905. (**D**) Genes of glycolysis/gluconeogenesis as annotated in the Kyoto Encyclopedia of Genes and Genomes (KEGG) pathway database show age-associated transcriptional decrease. Y-axis represented normalized read counts transformed by Log2. (41 genes used; see ***Supplementary file 4***; wilcoxon signed rank test, p=0.01). RNA-seq and genotype as in (**C**). (**E**) ChIP-qPCR validation. Genomic structures were shown according to the Flybase annotation (www.flybase.org). Along the gene of indicated genotype, a, b, c, and d were underlined, denoting the sites for PCR amplification. Color codes represented ATG (the translation start site), CDS (coding sequence), 5'UTR, and 3'UTR. Different primer sets confirmed that H3K27me3 modification was increased in aged WT and further increased in *utx* null mutants. H3K27me3 was reduced in PRC2 mutants as compared to WT, with the ratio between mutants and WT smaller than 1 (mean ±SD of three biological repeats; student *t*-test *p<0.05,

*Figure 5 continued on next page*

*Figure 5 continued*

**p<0.01; also see *Figure 5—figure supplement 1D*). ChIP-qPCR was from muscle tissues of 3d- and 30d-old male flies. Genotypes: WT: 5905. PRC2 mutant: *Pcl*[c421]/+; *Su(z)12*[c253]/+. *utx*[c499/c499]. (F–H) qRT-PCR analysis confirms that *Tpi* and *Pgi* of glycolytic genes have a decrease with age (F) but become increased in both PRC2 (G) and PRC1 mutants (H). (mean ±SD of three biological repeats; student *t*-test). qRT-PCR was from muscle tissues of 3d- and 30d-old male flies. Genotypes: WT: 5905. *Pcl*[c421]/+; *Su(z)12*[c253]/+. *Su(z)2*[c433]/+. (I) H3K27me3 increases in *utx* deficiency. H3K27me3 western blot (top) and quantification (bottom). (for H3K27me3 quantification: mean ±SD of three biological repeats; student *t*-test). Western blot was from muscle tissues of 30d-old male flies. Genotypes: WT: 5905. *utx*[c499/c499]. (J) *utx* null animals are short-lived. (for lifespan assay: 25°C; n ≥ 200 per genotype for curve; log-rank test). Genotypes as in (I). (K) qRT-PCR analysis indicates a further decrease of glycolytic genes in *utx*-deficient animals. (mean ±SD of three biological repeats; student *t*-test). qRT-PCR was from muscle tissues of 30d-old male flies. Genotypes as in (I).
DOI: https://doi.org/10.7554/eLife.35368.010

The following figure supplement is available for figure 5:

**Figure supplement 1.** Transcriptomics links H3K27me3 dynamics to the regulation of glycolytic genes.
DOI: https://doi.org/10.7554/eLife.35368.011

## Untargeted metabolomics shows enhanced glycolysis in long-lived PRC2 mutants

To further investigate the mechanism by which epigenetic changes regulate aging, we conducted another unbiased study by profiling cellular metabolites using LC-MS-based untargeted metabolomics (*Patti et al., 2012*). In total, 151 metabolites were identified. This experiment demonstrated changes in the levels of metabolites during aging of WT flies, and between WT and mutants (*Figure 6A and C*, *Figure 6—figure supplement 1A*). Pathway analysis revealed that glycolysis among additional metabolic processes was significantly enriched (*Figure 6B and D*). In particular, lactate, a specific indicator for anaerobic glycolysis (*Schurr and Payne, 2007*), was significantly decreased during normal aging in WT animals but became elevated in PRC2 mutants (*Figure 6E*). Of note, metabolites related to citric acid cycle remained unchanged between WT and mutants (*Figure 6—figure supplement 1B*). Combined, this evidence reveals a metabolic profile reflective of modulated glycolysis, being decreased during normal aging but increased in long-lived PRC2 mutants. Thus, this unbiased study using metabolomic approach showed modulation in the level of lactate and glycolysis during aging and in PRC2 mutants, in good congruence with the conclusion from the studies of epigenomic and transcriptional alterations (*Figure 5—figure supplement 1C and D*).

## Metabolic flux shows roles of PRC2 mutants in reversing glycolytic decline in aging

To provide quantitative insights into the changes in glucose metabolism with age and in PRC2 mutants, we devised a metabolic flux assay using $^{13}C$-labeled glucose as a tracer in adult aging flies. For young (3d) and aged (25d) animals, fly diet was switched from $^{12}C$-glucose to $^{13}C$-glucose, such that the glucose metabolism can be traced by measuring the $^{13}C$-labeled metabolites (*Figure 6F*, *Figure 6—figure supplement 2A*). After 5 days of feeding, $^{13}C$-labeled glucose accounted for more than 93% of the total glucose pool (*Figure 6—figure supplement 2B*), suggesting a near-complete replacement. This experiment demonstrated that while the level of labeled glucose was similar (*Figure 6—figure supplement 2C*), specific metabolites (*Figure 6G*) and the glycolytic pathway as a whole (*Figure 6H*) exhibited an age-associated decline; strikingly, the extent of decline was partially reversed by PRC2-deficiency. This finding, together with above results obtained by untargeted metabolomics, provides definitive evidence that glycolysis is modulated by aging. Since metabolites of the citric acid cycle are mix lineages from glycolysis, glutamate, and fatty acids, each metabolite may contain multiple $^{13}C$-labeled isotopologues via glycolysis-derived reactions as well as recurrent cycling (*Figure 6F*). Thus, we analyzed all isotopologues pertaining to each metabolites (*Figure 6—figure supplement 2D*) and their summed intensity (*Figure 6—figure supplement 2E*). Our data indicated no consistent change of the citric acid cycle with age and in PRC2-deficiency, which was in line with constant levels of metabolites as above measured by untargeted metabolomics (see *Figure 6—figure supplement 1B*).

These data combined establish a regulatory event from epigenetic alterations to transcriptional and metabolic modulations impacting the manner by which glucose can be metabolized during aging and in PRC2 mutants. While natural aging couples with to a metabolic decline including a

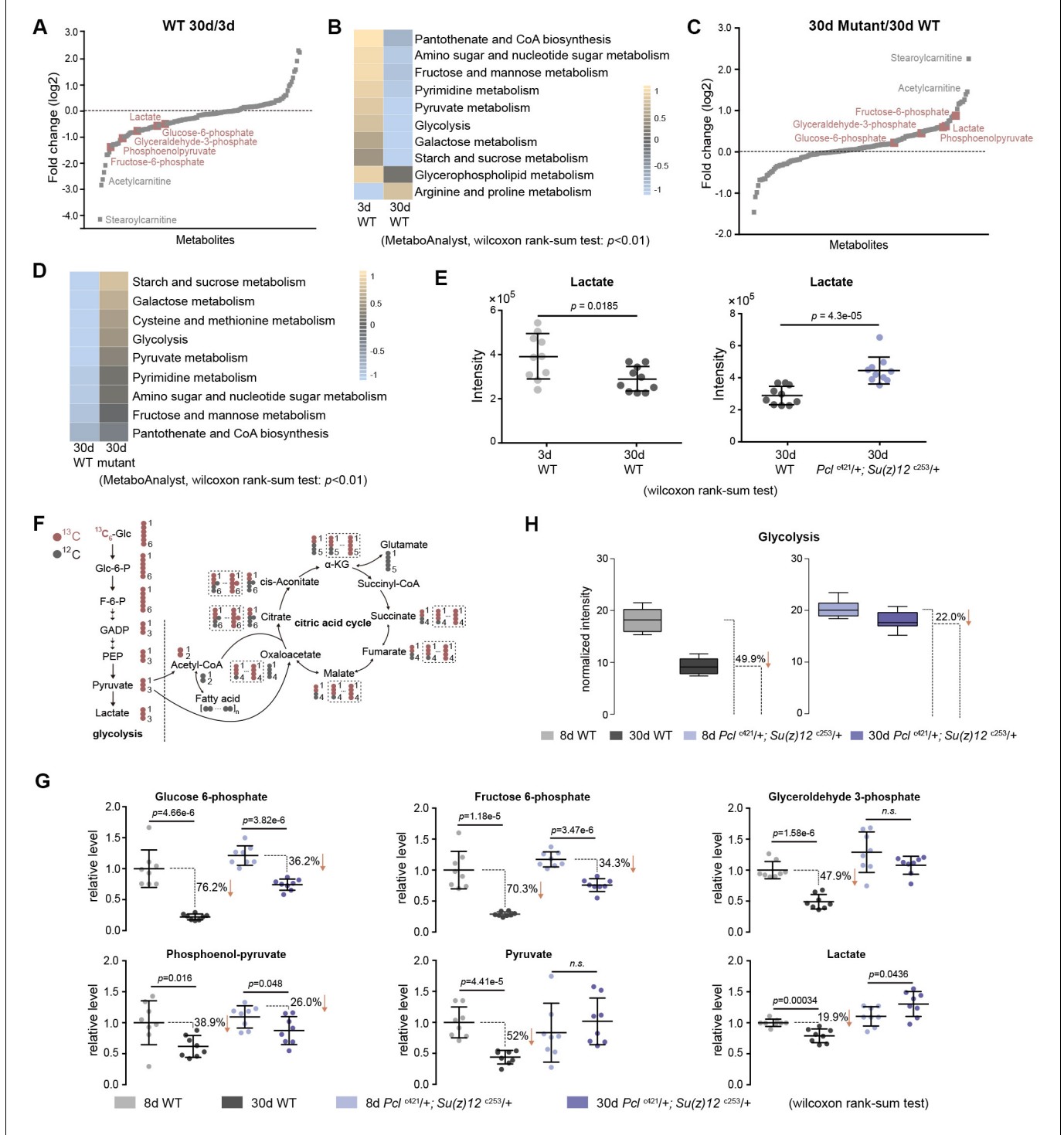

**Figure 6.** Metabolomics shows the effect of PRC2 mutants in reversing glycolytic decline in aging. (**A**) Untargeted metabolics identifies that metabolites of the glycolytic pathway (red) are decreased with age. For each metabolite, the median from 10 biological repeats was used to calculate the relative fold change between WT and mutants. Untargeted metabolics was from head tissues of 3d- and 30d-old male flies. Genotype: 5905. (**B**) Pathway enrichment analysis reveals glycolysis being the biological processes significantly enriched with age. Pathway enrichment analysis was executed using MetaboAnalyst. (Pathway enrichment criteria: wilcoxon rank-sum test, cutoff p<0.01; see Materials and methods for algorithm). Genotype as in (**A**). (**C**) Untargeted metabolics identifies that metabolites of the glycolytic pathway (red) are increased in PRC2 mutants. Untargeted metabolics and analysis as in (**A**). Genotypes: WT: 5905. Mut: $Pcl^{c421}/+$; $Su(z)12^{c253}/+$. (**D**) Pathway enrichment analysis reveals glycolysis being the biological processes significantly enriched. (Pathway enrichment criteria: wilcoxon rank-sum test, cutoff p<0.01; see method for algorithm). Genotype as in (**C**). (**E**) Lactate is

*Figure 6 continued on next page*

*Figure 6 continued*

decreased with age (left panel) but increased in PRC2 mutants (right panel). (mean ±SD of 10 biological repeats; wilcoxon rank-sum test). Genotypes: WT: 5905. *Pcl*<sup>c421</sup>/+; *Su(z)12*<sup>c253</sup>/+. (F) Schematic representation of the glucose flux via glycolysis (left) and citric acid cycle (right). $^{13}C$ and $^{12}C$ were indicated for specific metabolites. Dashed rectangles encircled isotopologues pertaining to specific metabolites that contained different number of $^{13}C$. (G) Decline of glycolytic metabolites during aging is partially rescued by PRC2 deficiency. For specific metabolite, the median from eight biological repeats was used to calculate the ratio between 8d and 30d of age. (mean ±SD of eight biological repeats; wilcoxon rank-sum test). Metabolomics was from head tissues of 8d- and 30d-old male flies. Genotypes: WT: 5905. *Pcl*<sup>c421</sup>/+; *Su(z)12*<sup>c253</sup>/+. (H) Age-associated decline of glycolysis is diminished by PRC2 deficiency. $^{13}C$-traced glycolytic metabolites were used to determine the level of glycolytic pathway as a whole. In 30d WT, 50.1% of glycolysis remained as relative to its level at 8d of age; in contrast, in PRC2 mutants at 30d of age, 78% of glycolysis remained. (mean ±SD of eight biological repeat; see method for calculation). Metabolomics and genotypes as in (G).

DOI: https://doi.org/10.7554/eLife.35368.012

The following figure supplements are available for figure 6:

**Figure supplement 1.** Untargeted metabolomics.
DOI: https://doi.org/10.7554/eLife.35368.013
**Figure supplement 2.** Metabolic glucose flux experiment.
DOI: https://doi.org/10.7554/eLife.35368.014

reduction of glycolysis, PRC2 deficiency relatively maintains the level of glycolysis as compared to age-matched WT animals.

## Stimulation of glycolysis promotes energy, redox state and adult fitness

We next characterized specific metabolites in the glycolytic pathway. While anaerobic glycolysis yields ATP only, glycolysis at the step of pyruvate formation has a net outcome of both ATP and NADH, a reduced form of nicotinamide adenine dinucleotide ($NAD^+$) (see *Figure 5—figure supplement 1C*). While the overall glucose levels between WT and mutants had no significant difference (*Figure 7A*), pyruvate was increased in mutants (*Figure 7A*). On the other hand, the ratio of $NADPH/NADP^+$, an indicator of the PPP, foliate metabolism, and malic enzyme (*Fan et al., 2014*) was even slightly decreased in mutants (*Figure 7B*). In parallel, we also examined the glutathione, an important indicator of the cellular redox state (*Jones, 2006*). Our data showed that ATP, NADH/$NAD^+$, and GSH/(GSH +GSSG) ratio were decreased with age but increased in mutants (*Figure 7C and D*). Combined, analysis of specific metabolites demonstrates a metabolic decline associated with normal aging, and that PRC2 deficiency promotes ATP and cellular redox potential as compared to age-matched WT controls.

We then asked how this ability of enhanced glycolysis in PRC2 mutants was translated into fitness and stress resistance in aging. To illuminate this, we characterized age-related phenotypes. Analysis of climbing of young (3d) and aged (30d) animals demonstrated that although WT and mutants at 3d behaved similarly, PRC2 mutants at 30d of age had better locomotion (*Figure 7E*). To examine the sensitivity to oxidative stress, we utilized hydrogen peroxide ($H_2O_2$), a source of reactive oxygen species (ROS). We observed substantially enhanced resistance to oxidation in mutants (*Figure 7F*). Elevated energy and NADH/$NAD^+$ ratio in PRC2 mutants might underlie improved motility and resistance to ROS, respectively. These data implicate that PRC2-deficiency couples longevity and metabolism with enhanced ability to handle stress, a property reflecting improved adult fitness.

## Perturbing glycolysis diminishes longevity traits in PRC2 mutants

We subsequently determined whether perturbing glycolysis could diminish the lifespan benefits of PRC2 mutants. In PRC2 mutants, *Tpi* and *Pgi* were two genes exhibiting a concerted decrease in H3K27me3 modification (see *Figure 5E*, *Figure 5—figure supplement 1D*) and a corresponding increase in transcription (see *Figure 5G*, *Figure 5—figure supplement 1C*); thus *Tpi* and *Pgi* might be important targets of PRC2. In light of this, we generated a null mutation of the *Tpi* gene by CRISPR/Cas9, *Tpi*<sup>c511</sup> (*Supplementary file 2*). *Tpi* gene, encoding triosephosphate isomerase, is a central glycolytic enzyme (*Bar-Even et al., 2012*). Given homozygous *Tpi*<sup>c511</sup> had a pre-adult lethality, we combined *Tpi*<sup>c511</sup> heterozygote with *Pcl*<sup>c421</sup> *Su(z)12*<sup>c253</sup>, and evaluated the adult phenotypes by using the triple mutants. Control studies indicated that the *Tpi*<sup>c511</sup> heterozygote alone retained normal lifespan (*Figure 8—figure supplement 1A*), oxidative stress (*Figure 8—figure supplement*

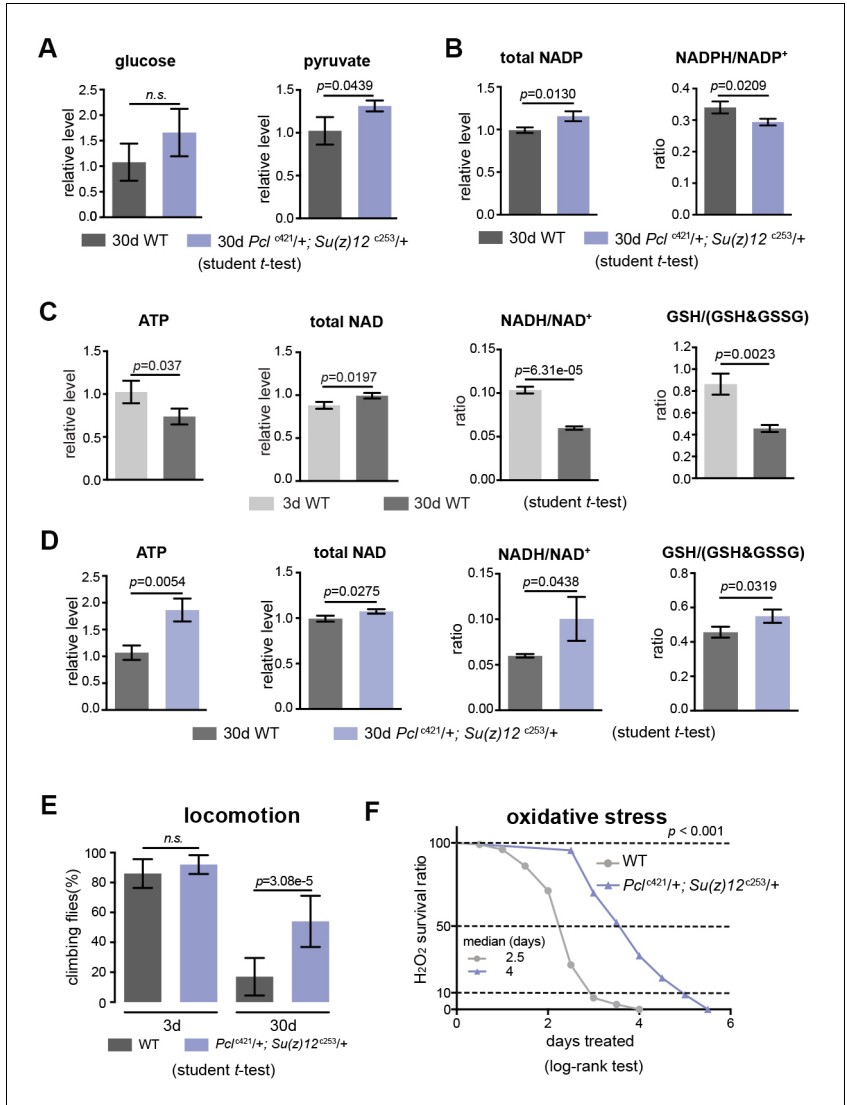

**Figure 7.** PRC2 mutants couple enhanced glycolysis with improved adult fitness. (**A**) Glucose content shows a slight increase in PRC2 mutants, but this increase is not statistically significant (left panel). Pyruvate, one key end product of glycolysis, is increased (right panel). (mean ±SD of three biological repeats with 10 flies for each measurements; student *t*-test; *n.s.*: not significant). Test was from muscle tissues of 30d-old male flies. Genotypes: WT: 5905. Mut: *Pcl*$^{c421}$/+; *Su(z)12* $^{c253}$/+. (**B**) The ratio of NADPH/NADP$^+$, indicator of the PPP, foliate metabolism, and malic enzyme is decreased in mutants. Test, statistics, and genotype as in (**A**). (**C**) ATP and cellular redox levels are decreased with age. (mean ±SD of three biological repeats with 10 flies for each measurements; student *t*-test). Test was from muscle tissues of 3d- and 30d-old male flies. Genotype: 5905. (**D**) ATP and cellular redox levels are increased in PRC2 deficient animals. Test, statistics, and genotype as in (**A**). (**E**) and (**F**) Analysis of adult phenotypes reveals that PRC2 mutants have a healthy lifespan. Climbing assay exhibited that, whereas WT and mutants behaved similarly at 3d, with age, mutants had better climbing, reflective of improved mobility (**E**). Mutants had enhanced resistance to oxidation (**F**). (for climbing assay: mean ±SD of 10 biological repeats with 10 flies for each repeats; student *t*-test; for oxidation tests: 25°C; n = 100 per genotype for curve; log-rank test). Genotypes as in (**A**).

DOI: https://doi.org/10.7554/eLife.35368.015

1B), locomotion (*Figure 8—figure supplement 1C*), and comparable levels of ATP and NADH/NAD$^+$ ratio (*Figure 8—figure supplement 1D*). Lowering *Tpi* in PRC2 mutants slightly decreased the expression of *Tpi* gene (*Figure 8—figure supplement 1E*) and diminished the extent by which glycolysis could be stimulated as having relatively reduced ATP and NADH/NAD$^+$ ratio (*Figure 8D*).

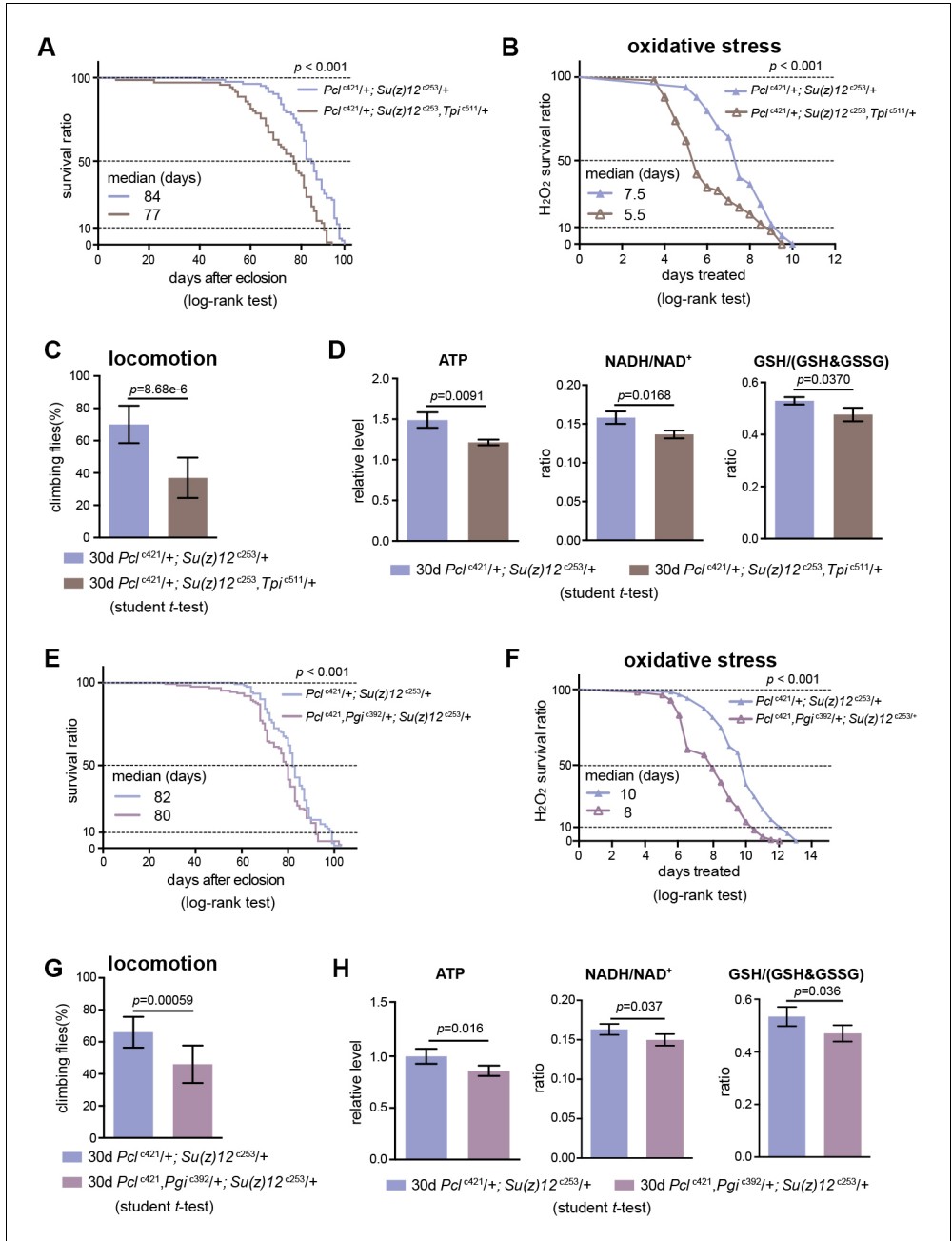

**Figure 8.** Perturbing glycolysis diminishes longevity benefits in PRC2 mutants. (**A**) *Tpi* deficiency significantly diminishes the longevity phenotype of PRC2 trans-heterozygous double mutants. (for lifespan assay: 25°C; n > 200 per genotype; log-rank test). Genotypes: $Pcl^{c421}/+$; $Su(z)12^{c253}/+$. $Pcl^{c421}/+$; $Su(z)12^{c253}$, $Tpi^{c511}/+$. (**B**) and (**C**) Oxidation stress test (**B**) and climbing (**C**) show that *Tpi* deficiency partially diminishes the lifespan-benefits of PRC2 mutants. (for oxidation assay: 25°C; n = 100 per genotype for curve; log-rank test; for climbing assay: mean ±SD of 10 biological repeats with 10 flies for each repeat; student *t*-test). Genotypes as in (**A**). (**D**) *Tpi* deficiency reduces glycolysis of PRC2 mutants. Analysis of specific metabolites revealed a decrease of ATP (left panel), ratio of NADH/NAD⁺ (middle panel), and ratio of GSH/(GSH +GSSG) (right panel). (mean ±SD of three biological repeats; student *t*-test). Metabolite analysis was from muscle tissues of 30d-old male flies. Genotypes as in (**A**). (**E–H**) *Pgi* deficiency diminishes the aging benefits mediated by PRC2 mutants, including lifespan (**E**), oxidative stress (**F**), locomotion (**G**), and glycolysis (**H**). Genotypes: $Pcl^{c421}/+$; $Su(z)12^{c253}/+$. $Pcl^{c421}$, $Pgi^{c392}/+$; $Su(z)12^{c253}/+$.

DOI: https://doi.org/10.7554/eLife.35368.016

The following figure supplement is available for figure 8:

*Figure 8 continued on next page*

*Figure 8 continued*

**Figure supplement 1.** *Tpi*- and *Pgi*-deficient flies have normal adult phenotypes.
DOI: https://doi.org/10.7554/eLife.35368.017

Analysis of lifespan exhibited that *Tpi* deficiency partially mitigated the longevity phenotype in PRC2 mutants (*Figure 8A*). Traits related to adult fitness, including resistance to ROS and climbing mobility, were also attenuated by *Tpi* deficiency (*Figure 8B and C*). To consolidate this result, we generated a null mutation for *Pgi* by CRISPR/Cas9, *Pgi*$^{c392}$ (*Supplementary file 2*). Consistently, while *Pgi*$^{c392}$ heterozygote alone had minimal effect on aging (*Figure 8—figure supplement 1F–1I*), triple mutants combining *Pgi*$^{c392}$ heterozygote with *Pcl*$^{c421}$ *Su(z)12*$^{c253}$ diminished the pro-lifespan benefits mediated by PRC2-deficiency (*Figure 8E–8H*, *Figure 8—figure supplement 1J*). These data implicate that perturbing glycolysis by genetic mutation diminishes longevity traits in PRC2 Mutants. Given the effect is partial, we cannot rule out the possibility that other mechanisms might be involved.

## Transgenic increase of glycolytic genes promotes healthy lifespan

Our data thus far have shown the effect of glycolysis in PRC2-dependent life-benefits. Furthermore, we investigated whether upregulating glycolytic genes in WT animals could be sufficient to extend lifespan. To address this, we increased the gene dosage of *Tpi* and *Pgi* via genomic transgenes, which expressed these genes under their endogenous regulatory elements (*Figure 9—figure supplement 1A and B*). Western blot analysis manifested that both Tpi and Pgi proteins had a modest decrease with age, but became upregulated upon PRC2-deficiency, an outcome consistent with the alteration in the level of H3K27me3 (*Figure 9A–9D*). We then interrogated the effects of *Tpi* and *Pgi* transgenes on WT flies. While single transgenes had modest phenotypes, animals combining both *Tpi* and *Pgi* transgenes stimulated glycolysis, as shown by elevated pyruvate, ATP, and NADH/NAD$^+$ ratio as compared to age-matched WT (*Figure 9E*). Accordingly, adult aging phenotypes, including lifespan (*Figure 9F*), locomotion (*Figure 9G*), and resistance to oxidative stress (*Figure 9H*), were improved.

Taken together, these data suggest that upregulation of glycolytic genes alone recapitulates anti-aging features of PRC2 mutants. This finding, combined with data obtained from H3K27me3 dynamics and consequently its effects on gene expression and metabolism, underscores the mechanistic link between epigenetic, transcriptional, and metabolic processes in aging, further heightening the role of glycolysis in promotion of metabolic health and longevity.

## Discussion

How changes in epigenomics, transcription, and metabolomics are integrated into the regulation of adult lifespan is a long-standing question in the biology of aging. Our study provides a new paradigm by which epigenetic drift of H3K27me3, a highly conserved histone mark, links glycolysis to healthy longevity (*Figure 9I*). Though H3K27me3 levels have been shown to increase with age in mammals (*Liu et al., 2013*; *Sun et al., 2014*) and flies (current study), the epigenomic feature of such changes is unknown. Our quantitative assessment shows that aging couples with a decline of fidelity in H3K27me3 modification, as demonstrated by a genome-wide drift of this repressive mark. Concomitant to this drift, glycolytic genes are downregulated, which in turn causes a reduction of glycolysis and glucose metabolism. Importantly, the age-delaying effect of PRCs-mutation likely arises from its ability to slow down or reverse this trend. Therefore, we suggest that epigenetic drift of H3K27me3, enhanced by its age-modulated increase, is one of the molecular mechanisms that drive the progression of aging. Expression of *Tpi* and *Pgi* of glycolytic genes is inherently regulated by the alterations of H3K27me3 levels, as shown by their corresponding changes during natural aging, in *utx* mutants as well as in PRCs-deficiency. We propose that H3K27me3 dynamics may elicit changes in local chromatin environment—especially inter-peak regions that are potentially more sensitive to the gain or loss of the epigenetic mark—thus enabling fine-tuning of target gene expression.

It is unclear how aging might promote the drift in H3K27me3 modification. One possible reason is age-associated DNA damage that reduces the fidelity of epigenetic markings. It has been well-

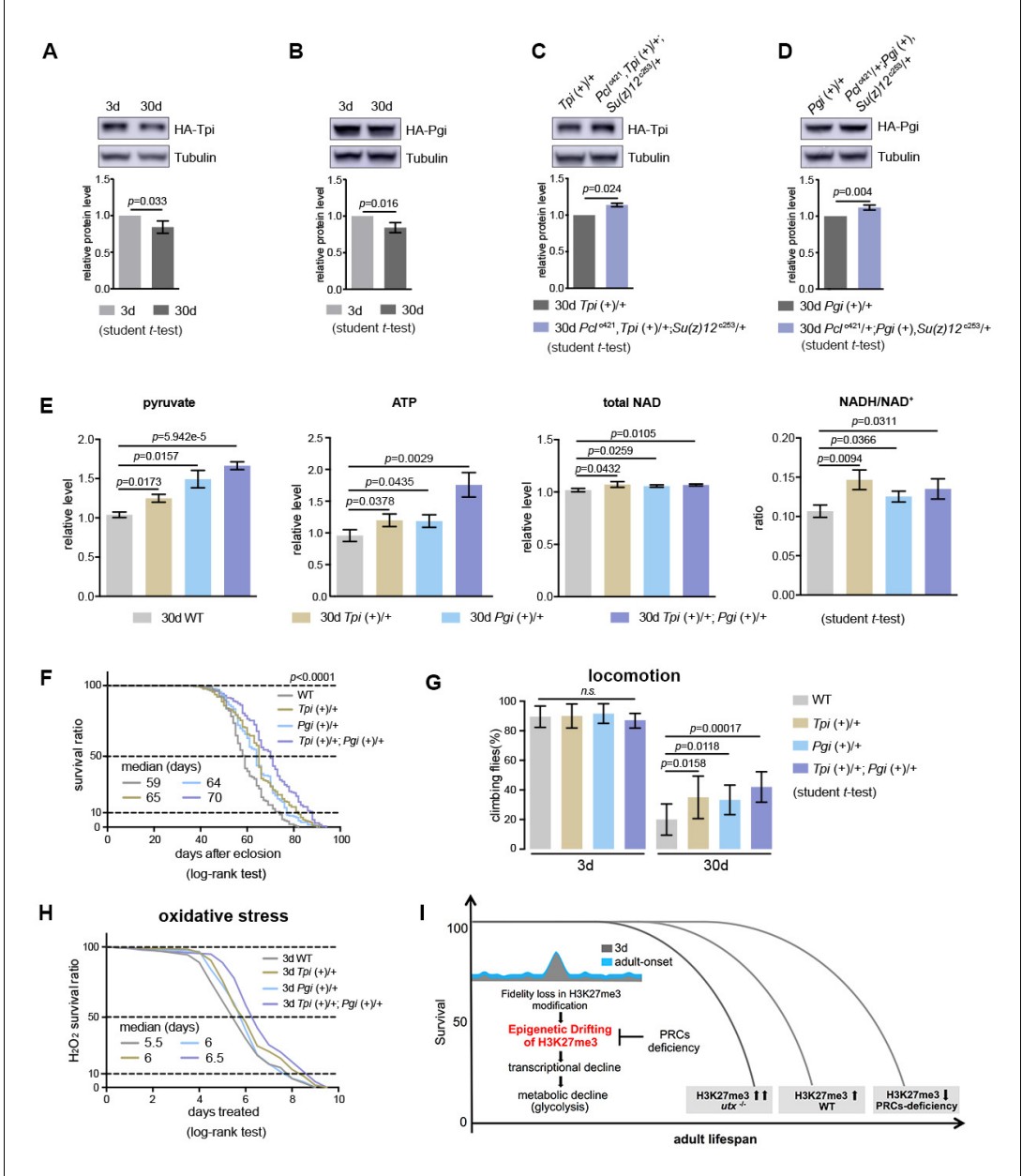

**Figure 9.** Transgenic increase of glycolytic genes suffices to elevate glycolysis and healthy lifespan. (A) Tpi protein western blot (top) and quantification (bottom) shows a decrease with age. (mean ±SD of three biological repeats; student *t*-test). Western blot was from 3d- and 30d-old male flies. Genotype: *Tpi* (+)/+. (B) Pgi protein western blot (top) and quantification (bottom) shows a decrease with age. Statistics and Western blot as in (A). Genotype: *Pgi* (+)/+. (C) Tpi protein western blot (top) and quantification (bottom) shows an increase in PRC2 mutants. Statistics and western blot as in (A). Genotype: *Tpi* (+)/+. *Pcl*^c421^, *Tpi* (+)/+; *Su(z)12* ^c253^/+. (D) Pgi protein western blot (top) and quantification (bottom) shows an increase in PRC2 mutants. Statistics and western blot as in (A). Genotype: *Pgi* (+)/+. *Pcl*^c421^/+; *Pgi* (+), *Su(z)12* ^c253^/+. (E) Transgenic increase of glycolytic genes stimulates glycolysis, as shown by elevated pyruvate, ATP, and NADH/NAD$^+$ ratio compared to age-matched WT (mean ±SD of three biological repeats; student *t*-test). Metabolite analysis was from muscle tissues of 30d-old male flies. Genotypes: WT: 5905. *Tpi* (+)/+. *Pgi* (+)/+. *Tpi* (+)/+; *Pgi* (+)/+. (F–H) Transgenic increase of glycolytic genes promotes adult fitness, including lifespan (F), locomotion (G), and resistance to oxidative stress (H). (for lifespan assay: 25˚C; n ≥ 200 per genotype for curve; log-rank test; for climbing assay: mean ±SD of 10 biological repeats with 10 flies for each repeat; student *t*-test; for oxidation assay: 25˚C; n = 100 per genotype for curve; log-rank test). Genotypes as in (E). (I) Model. Adult-onset fidelity loss results in epigenetic drift of H3K27me3. Over a chronic timescale, the drifting of H3K27me3 induces transcriptional and metabolic decline including a reduction of glycolysis. Effects of PRC2-deficiency in life-extension can be at least in part attributed to the effect in stimulation of glycolysis, thereby maintaining metabolic health and longevity. Adult lifespan is inherently modulated by the alterations in the levels of H3K27me3, as shown by the corresponding change during natural aging, in *utx* mutants as well as in PRCs-deficiency.

DOI: https://doi.org/10.7554/eLife.35368.018

*Figure 9 continued on next page*

*Figure 9 continued*

The following figure supplement is available for figure 9:

**Figure supplement 1.** Genomic transgenes for *Tpi* and *Pgi*.

DOI: https://doi.org/10.7554/eLife.35368.019

established that aging is associated with the accumulation of DNA damage (*Lu et al., 2004*). In *Bombyx mori*, UV-C irradiation can induce the increase of H3K27me3 mediated by PRC2 (*Li et al., 2014*). In *Neurospora crassa*, on the other hand, the increase and redistribution of H3K27me3 can be induced by the loss of H3K9 methyltransferase complex (*Basenko et al., 2015*). Thus, dysregulation of H3K27me3 may be due to feed-forward interactions between DNA damage and changes in the levels of complexes that control epigenetic markings during aging.

Metabolic control has emerged as a critical player to determine the aging outcome. Previous data implicates that impairing glycolysis can extend lifespan perhaps by triggering compensatory processes (*Schulz et al., 2007*). Quantitative proteomic analysis in mouse heart shows that aging correlates with an increased abundance of proteins in glycolysis, and that caloric restriction and rapamycin treatment known to extend longevity can reverse this trend (*Dai et al., 2014*). D-Glucosamine supplementation that impairs glucose metabolism can extend lifespan in *C. elegans* and mouse (*Weimer et al., 2014*). However, rat models supplemented with 2-deoxyglucose, a known anti-glycolytic compound, are instead short-lived (*Minor et al., 2010*). Moreover, glucose-enriched diet can prevent pathophysiological decline and early death of telomere dysfunctional mice by stimulating glucose metabolism including glycolysis (*Missios et al., 2014*). Importantly, recent evidence from studies on a wide range of species suggests that a balanced diet from protein to carbohydrate and their interactive effects are key elements for healthy longevity. Mouse dietary studies have demonstrated that animals fed on diets that are low in protein and high in carbohydrate enjoy longer life with lower blood pressure, improved glucose tolerance and lipid profiles (*Solon-Biet et al., 2014*). This is also supported by human data on the deleterious effects of diets with high protein and low-carbohydrate (*Floegel and Pischon, 2012*; *Lagiou et al., 2012*). Hence, the impact of glucose metabolism including glycolysis on natural aging might be poised to integrate age-associated physiology; the precise outcome must be rather complicated, which could be influenced not only by diverse spatiotemporal need in normal or pathological conditions, but also upon newly evolved biological activities correlated with the increased complexity from nematode to *Drosophila* and mammals. It is important to note that since our study is based on *Drosophila* somatic tissues that are primarily post-mitotic cells in adult animals, it is possible that the effects are more dominant in the fly model than others. Our findings in *Drosophila* propose that, during natural aging, the ability to maintain glycolysis is an alternative avenue to ensure a healthy level of physical capability by delaying the bioenergetic decline with advanced age as which otherwise might form a vicious cycle from decrease in physical activity to hypometabolism, and perhaps disease susceptibility.

Aging is a complex process that can be regulated by a network of multiple mechanisms. It has been well-established that enhancing $NAD^+$ biogenesis promotes healthy lifespan (*Anderson et al., 2002*; *Balan et al., 2008*; *Mills et al., 2016*). As noted, supplementation of $NAD^+$ precursors profoundly elevates energy metabolism by increasing the expression of genes in the TCA cycle as well as glycolysis in *C. elegans* (*Mouchiroud et al., 2013*), and promotes glucose metabolism with increased flux through pentose phosphate and glycolytic pathways in mice on a high-fat diet (*Mitchell et al., 2018*). Therefore, it would be interesting to test whether the life-benefit effects of $NAD^+$ might be through at least in part by the activation of glycolysis. Intriguingly, while declining intracellular $NAD^+$ and thus increased $NADH/NAD^+$ ratio correlate with aging (*Zhu et al., 2015*), our experiments in *Drosophila* demonstrate that increased ratios of GSH/(GSH +GSSG) and $NADH/NAD^+$ due to enhanced glycolytic activities may provide a simple but effective way to retard aging. The oxidative phosphorylation, although produces more ATP than glycolysis, can yield intracellular ROS. The accumulation of ROS is the leading proposed cause of decline in cellular function and integrity in aging (*Balaban et al., 2005*). Thus, modulating H3K27me3 may reprogram bioenergetic decline during aging, which in effect reduces cellular damage and deterioration. Importantly, mammalian glycolytic genes have also been shown as PRCs targets (*Brookes et al., 2012*). Future investigations, including in-depth comparative analysis of PRCs and glycolytic pathway in the aging

process in both flies and humans, may harness common operative mechanisms that modulate metabolic homeostasis and healthy longevity. Given the reversible nature of epigenetic pathways, this study proffers a tempting strategy against age-associated physiological decline and disease.

# Materials and methods

**Key resources table**

| Reagent type (species) or resource | Designation | Source or reference | Identifiers | Additional information |
|---|---|---|---|---|
| Cell line (*D. melanogaster*) | S2R+ | This paper | FLYB:FBtc0000150 | |
| Antibody | H3K27me3 (rabbit polyclonal) | Millipore | Cat#07–449; RRID:AB_310624 | |
| Antibody | H3K27me2 (rabbit polyclonal) | Abcam | Cat#ab24684; RRID:AB_448222 | |
| Antibody | H3K27me (rabbit polyclonal) | Millipore | Cat#07–448 | |
| Antibody | H3K27ac (rabbit polyclonal) | Abcam | Cat#ab4729; RRID:AB_2118291 | |
| Antibody | H3K9me3 (rabbit polyclonal) | Abcam | Cat#ab8898; RRID:AB_306848 | |
| Antibody | H3K9ac (rabbit polyclonal) | Millipore | Cat#06–942; RRID:AB_310308 | |
| Antibody | H3K23ac (rabbit polyclonal) | Millipore | Cat#07–355; RRID:AB_310546 | |
| Antibody | H3K4me3 (rabbit polyclonal) | Millipore | Cat#07–473; RRID:AB_1977252 | |
| Antibody | H3K4me2 (rabbit polyclonal) | Millipore | Cat#07–030; RRID:AB_310342 | |
| Antibody | H3K4me (rabbit polyclonal) | Millipore | Cat#07–436; RRID:AB_310614 | |
| Antibody | H3K36ac (rabbit polyclonal) | Millipore | Cat#07–540 | |
| Antibody | H3K36me3 (rabbit polyclonal) | Abcam | Cat#ab9050; RRID:AB_306966 | |
| Antibody | H3K14ac (rabbit polyclonal) | Millipore | Cat#07–353; RRID:AB_310545 | |
| Antibody | H3K18ac (rabbit polyclonal) | Abcam | Cat#ab1191; RRID:AB_298692 | |
| Antibody | H4K20me3 (rabbit polyclonal) | Abcam | Cat#ab9053 | |
| Antibody | H3 (goat polyclonal) | Abcam | Cat#Ab12079 | |
| Antibody | HA (rabbit monoclonal) | Cell signaling technology | Cat#3724 | |
| Antibody | Tubulin (rabbit polyclonal) | Medical and biological laboratories co | Cat#PM054 | |
| Antibody | Anti-Rabbit IgG | Sigma | Cat#A9169 | |
| Antibody | Anti-Mouse IgG | Sigma | Cat#A4416 | |
| Antibody | Anti-Goat IgG | Abcam | Cat#ab6471 | |
| Recombinant DNA reagent | pBID UAS vector: HA-Tpi | Addgene | 35198 | |
| Recombinant DNA reagent | pBID UAS vector: HA-Pgi | Addgene | 35198 | |
| Recombinant DNA reagent | pBID vector: HA-Tpi genomic | Addgene | 35190 | |

*Continued on next page*

*Continued*

| Reagent type (species) or resource | Designation | Source or reference | Identifiers | Additional information |
|---|---|---|---|---|
| Recombinant DNA reagent | pBID vector: HA-Pgi genomic | Addgene | 35190 | |
| Recombinant DNA reagent | pU6b-sgRNA | *Ren et al., 2013* | N/A | |
| Commercial assay or kit | TURBO DNA-free kit | ThermoFisher | Cat#AM1907 | |
| Commercial assay or kit | SuperSignal West Pico Chemilluminescent Subtrate | ThermoFisher | Cat#34078 | |
| Commercial assay or kit | Nuclear-Cytosol Extraction Kit | Aogma | Cat#9988 | |
| Commercial assay or kit | SYBR Select Master Mix | ThermoFisher | Cat#26161 | |
| Commercial assay or kit | SuperScript III First-strand synthesis system for RT-PCR | ThermoFisher | Cat#18080–051 | |
| Commercial assay or kit | ChIP-Grade Protein A/G Plus Agarose | ThermoFisher | Cat#26161 | |
| Commercial assay or kit | QIAquick PCR Purification Kit | QIAGEN | Cat#28106 | |
| Commercial assay or kit | Qubit dsDNA HS Assay Kit | ThermoFisher | Cat#Q32851 | |
| Commercial assay or kit | High Sentivity DNA Analysis Kits | Agilent Technologies | Cat#5067–4626 | |
| Commercial assay or kit | NEBNext DNA Library Prep Kit | New England Biolabs | Cat#E7370L | |
| Commercial assay or kit | NEBNext Poly(A) mRNA Magnetic Isolation Module | New England Biolabs | Cat#E7490L | |
| Commercial assay or kit | NEBNext Multiplex Oligos for Illumina | New England Biolabs | Cat#E7600S | |
| Commercial assay or kit | NEBNext Ultra RNA library Prep Kit for Illumina | New England Biolabs | Cat#E7530L | |
| Commercial assay or kit | RNase A/T1 Mix | ThermoFisher | Cat#EN0551 | |
| Commercial assay or kit | Glucose (HK) Assay Kit | Sigma | Cat#GAHK20 | |
| Commercial assay or kit | NAD/NADH Quantitation Kit | Sigma | Cat#MAK037 | |
| Commercial assay or kit | NADP/NADPH Quantitation Kit | Sigma | Cat#MAK038 | |
| Commercial assay or kit | ENLITEN ATP Assay System | Promega | Cat#FF2000 | |
| Commercial assay or kit | Pyruvate Assay Kit | Sigma | Cat#: MAK071 | |
| Commercial assay or kit | GSH/GSSG Ratio Detection Assay | Abcam | Cat#ab13881 | |
| Commercial assay or kit | Amicon Ultra 0.5 mL centrifugal filters | Sigma | Cat#Z677108-96EA | |
| Chemical compound, drug | Tris-buffer (1 mol/L, pH8.5) | Sangon Biotech | Cat#SD8141 | |
| Chemical compound, drug | TRIZol Rreagent | ThermoFisher | Cat#15596018 | |
| Chemical compound, drug | Chloroform | Sinopharm Chemical Reagent | Cat#10006818 | |
| Chemical compound, drug | Isopropanol | Sinopharm Chemical Reagent | Cat#10006818 | |
| Chemical compound, drug | DEPC-treated water | Invitrogen | Cat#46–2224 | |
| Chemical compound, drug | NuPAGE 12% Bis-Tris Gel | ThermoFisher | Cat#NP0342BOX | |
| Chemical compound, drug | PageRuler Prestained Protein Ladder | ThermoFisher | Cat#26616 | |

*Continued*

| Reagent type (species) or resource | Designation | Source or reference | Identifiers | Additional information |
|---|---|---|---|---|
| Chemical compound, drug | NuPAGE MOPS SDS Running Buffer | ThermoFisher | Cat#NP0001 | |
| Chemical compound, drug | Immobilon Transfer Membranes | Millipore | Cat#: IPVH00010 | |
| Chemical compound, drug | Glycine | beyotime | Cat#ST085-1000G | |
| Chemical compound, drug | 20 X PBS | Sangon Biotech | Cat#B548117-0500 | |
| Chemical compound, drug | 37% formaldehyde solution | Sigma | Cat#F1635 | |
| Chemical compound, drug | Protease inhibitor cocktail tablets | Roche | Cat#11697498001 | |
| Chemical compound, drug | RIPA buffer | Sigma | Cat#R-278–500 ML | |
| Chemical compound, drug | 5M NaCl | ThermoFisher | Cat#AM9760G | |
| Chemical compound, drug | 0.5M EDTA | ThermoFisher | Cat#15575–038 | |
| Chemical compound, drug | 1M Tris-HCl | ThermoFisher | Cat#15568–025 | |
| Chemical compound, drug | Triton X-100 | Sigma | Cat#T8787-50ML | |
| Chemical compound, drug | 10% SDS | Sangon Biotech | Cat#SD8118 | |
| Chemical compound, drug | RNase A/T1 Mix | ThermoFisher | Cat#EN0551 | |
| Chemical compound, drug | Sodium dicarbonate | Sigma | Cat#S5761 | |
| Chemical compound, drug | Proteinase K | Sangon Biotech | Cat#A100706 | |
| Chemical compound, drug | 3M Sodium Acetate pH5.5 | ThermoFisher | Cat#AM9740 | |
| Chemical compound, drug | Methyl viologen dichloride hydrate | Sigma | Cat#856177 | |
| Chemical compound, drug | Hydrogen peroxide 30% | Sinopharm Chemical Reagent | Cat#10011208 | |
| Chemical compound, drug | D(+)-Arabinose | Sangon Biotech | Cat#A600071-0100 | |
| Chemical compound, drug | D-glucose | Sangon Biotech | Cat#A100188-0500 | |
| Chemical compound, drug | Sodium hydroxide solution | Sigma | Cat#72068 | |
| Chemical compound, drug | Methanol | Honeywell | Cat#LC230-2.5HC | |
| Chemical compound, drug | Acetonitrile | Merck | Cat#1.00029.2500 | |
| Chemical compound, drug | Water | Honeywell | Cat#LC365-2.5HC | |
| Chemical compound, drug | Ammonium acetate | Sigma-Aldrich | Cat#73594–25 G-F | |
| Chemical compound, drug | Ammonium hydroxide | Sigma-Aldrich | Cat#44273–100 mL-F | |

*Continued on next page*

*Continued*

| Reagent type (species) or resource | Designation | Source or reference | Identifiers | Additional information |
|---|---|---|---|---|
| Chemical compound, drug | Sequencing Grade Modified Trypsin | Promega | Cat#V5111 | |
| Chemical compound, drug | Centrifugal Filters Ultracel 30K | Amicon | Cat#UFC30SV00 | |
| Chemical compound, drug | Water, LC/MS | J.T.Baker | Cat#9831–03 | |
| Chemical compound, drug | Formic acid, eluent additive for LC-MS | Sigma | Cat#56302–50 ml-f | |
| Chemical compound, drug | Acetonitrile,LC/MS,4L | J.T.Baker | Cat#9829–03 | |
| Chemical compound, drug | Ammonium bicarbonate | Sigma | Cat#09830–1 KG | |
| Chemical compound, drug | Urea | Sigma | Cat#U5128-500G | |
| Chemical compound, drug | Trizma hydrochloride | Sigma | Cat#T5941-100G | |
| Chemical compound, drug | Ammonium formate | Flucka | Cat#17843–50G | |
| Chemical compound, drug | Lys(6) SILAC yeast | Silantes | Cat#234CXX-SYK6-509-01 | |
| Chemical compound, drug | D-Glucose (U-$^{13}C_6$,99%) | Cambridge Isotope Laboratories, Inc. | Cat#110187-42-3 | |
| Software, algorithm | Bowtie2 | *Langmead and Salzberg (2012)* | http://bowtie-bio.sourceforge.net/bowtie2/index.shtml | |
| Software, algorithm | Samtools | *Li et al. (2009)* | http://samtools.sourceforge.net/; RRID:SCR_002105 | |
| Software, algorithm | deepTools | *Ramírez et al., 2014* | https://deeptools.github.io/ | |
| Software, algorithm | homer | *Heinz et al., 2010* | http://homer.ucsd.edu/homer/ngs/index.html; RRID:SCR_010881 | |
| Software, algorithm | bwtool | *Pohl and Beato, 2014* | https://users.dcc.uchile.cl/~peortega/bwtool/; RRID:SCR_003035 | |
| Software, algorithm | Galaxy | *Grüning et al., 2017* | https://github.com/bgruening/galaxytools; RRID:SCR_006281 | |
| Software, algorithm | Cistrome | *Liu et al., 2011* | http://cistrome.dfci.harvard.edu/ap/; RRID:SCR_000242 | |
| Software, algorithm | BEDTools | *Quinlan and Hall, 2010* | http://bedtools.readthedocs.io/; RRID:SCR_006646 | |
| Software, algorithm | J-circos | *An et al., 2015* | https://omictools.com/j-circos-tool; RRID:SCR_011798 | |
| Software, algorithm | VENNY | *Oliveros, 2007* | http://bioinfogp.cnb.csic.es/tools/venny/index.html | |
| Software, algorithm | R | *R Core Team, 2013* | https://www.r-project.org; RRID:SCR_001905 | |
| Software, algorithm | DAVID Bioinformatics Resources | *Huang et al., 2009b* | https://david.ncifcrf.gov/; RRID:SCR_001881 | |
| Software, algorithm | IGV | *Robinson et al., 2011* | http://software.broadinstitute.org/software/igv/; RRID:SCR_011793 | |
| Software, algorithm | STAR | *Dobin et al., 2013* | https://github.com/alexdobin/STAR; RRID:SCR_005622 | |

*Continued on next page*

*Continued*

| Reagent type (species) or resource | Designation | Source or reference | Identifiers | Additional information |
|---|---|---|---|---|
| Software, algorithm | HTSeq | *Anders et al., 2015* | http://www-huber.embl.de/HTSeq/; RRID:SCR_005514 | |
| Software, algorithm | DESeq | *Anders and Huber, 2010* | http://www-huber.embl.de/users/anders/DESeq/; RRID:SCR_000154 | |
| Software, algorithm | WGCNA | *Langfelder and Horvath, 2008* | https://labs.genetics.ucla.edu/horvath/htdocs/CoexpressionNetwork/Rpackages/WGCNA/; RRID:SCR_003302 | |
| Software, algorithm | XCMS | *Smith et al., 2006* | http://www.bioconductor.org/packages/release/bioc/html/xcms.html; RRID:SCR_015538 | |
| Software, algorithm | MetaboAnalyst 3.0 | *Xia et al., 2015* | http://www.metaboanalyst.ca/; RRID:SCR_015539 | |
| Software, algorithm | CAMERA | *Kuhl et al. (2012)* | https://bioconductor.org/packages/release/bioc/html/CAMERA.html; RRID:SCR_011924 | |
| Software, algorithm | PD1.4 | Thermo | https://www.thermofisher.com/order/catalog/product/IQLAAEGABSFAKJMAUH | |
| Software, algorithm | Pathways to PCDL (version B.07.00) | Agilent Technologies | | |
| Software, algorithm | PCDL Manager (version B.07.00) | Agilent Technologies | | |
| Software, algorithm | Profinder (version B.08.00) | Agilent Technologies | | |
| Software, algorithm | Prism | GraphPad | v6; RRID:SCR_002798 | |

## Fly culture

Flies were cultured in standard media at 25°C with 60% humidity in a 12 hr light and 12 hr dark cycle unless otherwise specified. Briefly, standard *Drosophila* food contains sucrose (36 g/L), maltose (38 g/L), yeast (22.5 g/L), agar (5.4 g/L), maizena (60 g/L), soybean flour (8.25 g/L), sodium benzoate (0.9 g/L), methyl-p-hydroxybenzoate: (0.225 g/L), propionic acid (6.18 ml/L), and ddH$_2$O to make up 1 L of the food. The WT line used was 5905 (FlyBase ID FBst0005905, $w^{1118}$). All fly lines used in this study have been backcrossed with 5905 for five consecutive generations for a uniform genetic background, to assure that phenotypes were not associated with any variation in background.

## Western blot assays for different histone modifications

Fifty heads or 30 muscles were harvested and immediately frozen in liquid nitrogen. Tissues were homogenized by Dounce Homogenizer. For total protein extract, tissues were lysed by RIPA buffer and then treated with five cycles of sonication with 30 s on and 30 s off (Bioruptor Pico, Belgium). Resulted histone extractions were loaded onto a NuPAGE 12% Bis-Tris Gel (Thermo Fisher Scientific, Carlsbad, CA, USA), and then transferred by electrophoresis to a polyvinylidene fluoride membrane (Millipore, Germany). The corresponding primary antibodies were diluted 1:1000 and incubated with the membranes at 4°C overnight, the HRP-labeled secondary antibodies were diluted 1:40000 and incubated for 1 hr at room temperature. The signals were detected using ECL (Thermo Fisher Scientific) by Amersham Imager 600 (GE Healthcare, Sweden). The signal intensity was quantified by ImageQuant TL (GE Healthcare). The relative normalization of histone modifications was normalized to the histone H3 signal. Three biological replicates were done for each experiment.

## CRIPSR/Cas9 mutagenesis

CRISPR/Cas9 mutagenesis was performed as previously described (*Ren et al., 2013*). Two sgRNA plasmids for target gene were injected into fly embryo. Single-fly-PCR assays were used to screen mutants. To do this, single fly was homogenized in 50 μL squashing buffer (10 mM Tris buffer (pH 8.5), 25 mM NaCl, 1 mM EDTA, 200 μg/mL Proteinase K), then incubated at 37℃ for 30 min, followed by 95℃, 10 min for inactivation. Screen primers for each target genes were listed in *Supplementary file 2*. The virgin females carrying the deletion were backcrossed into WT (5905) male flies for five consecutive generations for a uniform genetic background, to mitigate background effects.

## Genomic DNA transgene

Genomic DNA covering the Tpi and Pgi genes was prepared by PCR-amplification and fusion PCR. DNA was clone into the pBID vector (*Wang et al., 2012*). Fly microinjection and transgenesis followed standard procedures.

## Metabolic flux analysis using U-[$^{13}$C]-glucose

Metabolic flux analyses are described in more detail at Bio-protocol (*Cai et al., 2018*). 160 male flies for each genotype at 3d or 25d of age were used, with 20 flies per vial. Prior to the test, flies were pre-treated with 1% Agar media for 6 hr before transferred to the vials containing a small piece of Kimwipe filter paper pre-soaked in 1 mL of 10% U-$^{13}C_6$-glucose. Flies were treated for 3 days, and then transferred to new vials with fresh U-$^{13}C_6$-glucose for additional 2 days. Fly heads were dissected for subsequent metabolic analysis. For each genotype, eight biological repeats were conducted, with 20 heads for each repeats.

## Assays for adult phenotypes

### Lifespan assays

Adult male flies were collected at the day of eclosion and maintained at 20 flies per vial at 25℃ with 60% humidity and a 12 hr light/12 hr dark cycle. Flies were transferred to new vials every other day and scored for survival. The Prism software (Graphpad) was used to draw the survival curve and statistical analysis.

### Climbing ability assay

Ten male flies were transferred to an empty vial for 30 min adaption in dark. Flies were tapped to the bottom of the vial; the percentage of flies failed to climb up to a mark at 2 cm from the bottom within 5 s was scored. Ten biological replicates were done for each genotype at given age.

### Oxidative stress

One hundred male flies for each genotype at given age were used, with 20 flies per vial. Prior to the test, flies were pre-treated with 1% Agar media for 6 hr before transferred to the media containing oxidative radical species. For $H_2O_2$ treatment, flies were transferred to vials containing a small piece of Kimwipe filter paper pre-soaked in 1.5 ml of 10% glucose and 2% $H_2O_2$ (Sinopharm Chemical Reagent). Dead flies were scored every 12 hr, and the Prism software (Graphpad) was used to generate the survival curve and statistical analysis.

## Sequencing experiments

### ChIP-seq

Quantitative ChIP-seq analyses are described in more detail at Bio-protocol (*Niu et al., 2018*). For developmental stages, embryo (30 min-1 hr), larva (96 hr), and pupae (7d) were analyzed. For adult stage, 100 fly muscles and 200 fly heads at each time point were used. Samples were collected into a 1.5-mL microcentrifuge tube and immediately frozen in liquid nitrogen. Tissues were pulverized to fine powder using a liquid nitrogen cooled mortar and pestle. Resulted fine power was resuspended in 1.2 mL PBS; 32.4 μL of 37% formaldehyde was added for cross-linking at room temperature for 10 min. To quench formaldehyde, 60 μL of 2.5 M glycine was added. Centrifugation was using 5000 *g* for 5 min at 4℃. After removing the supernatant, pellet was washed twice with 1 ml PBS containing

proteinase inhibitor (Roche). This lysis step was performed at 4°C for more than 1 hr in RIPA buffer (Sigma) supplemented with proteinase inhibitor. The lysate was sonicated using Bioruptor Pico (Diagenode), for 15 cycles with 30 s on and 30 s off. Centrifugation was using 10,000 *g* for 10 min at 4°C. Supernatant was carefully transferred to a new tube without disturbing the debris. Pre-clear was performed using ChIP-Grade Protein A/G Plus Agarose (Thermo Fisher Scientific) at 4°C for 1 hr. For spike-in epigenome, mouse neuro-2a cells were prepared with the same procedures. Prior to mixing mouse epigenome with that of fly's, 5% from each preparations was transferred for quantification of DNA mass. To do this, RIPA buffer was added to a final volume of 120 µl, followed by 1 µl RNase A to remove the RNA at 37°C for 30 min. Samples were treated with 0.2 M NaCl (final concentration) at 65°C for 4 hr and 200 ng/µl proteinase K (final concentration) at 55°C for 2 hr. DNA was purified by PCR purification kit (Qiagen), and DNA mass was quantified by Qubit dsDNA HS assay kit (Thermo Fisher Scientific). According to the DNA mass, 5% of the mouse epigenome was added to the fly sample. 10% of sample was transferred to a new tube as ChIP input control. Immunoprecipitation was performed by incubating 3 µg antibody at 4°C for 5 hr with rotation. ChIP-Grade Protein A/G Plus Agarose was added. The sample was incubated overnight at 4°C with rotation. Beads were washed twice with ChIP Wash Buffer (150 mM NaCl, 2 mM EDTA (pH8.0), 20 mM Tris- HCl (pH8.0), 1% Triton X-100, and 0.1% SDS), and another two washes using ChIP Final Wash Buffer (500 mM NaCl, 2 mM EDTA (pH8.0), 20 mM Tris-HCl (pH8.0), 1% Triton X-100, and 0.1% SDS). The elution step was using 120 µL of ChIP Elution Buffer (100 mM NaHCO$_3$ and 1% SDS) by incubating at 65°C for 30 min. Resulted DNA was purified as above described, followed by RNase A treatment. 5–10 ng of DNA was used to generate sequencing library using NEB DNA library prep kit. The quality of libraries was checked by Bioanalyzer 2100 (Agilent). The quantification was performed by qRT-PCR with a reference to a standard library. The libraries were pooled together in equimolar amounts to a final 2 nM concentration. The normalized libraries were denatured with 0.1 M NaOH (Sigma). Pooled libraries were sequenced on the illumina Miseq/Next-seq platform with single end 100 bps.

### polyA-selected RNA-seq
Thirty muscles and fifty heads were dissected and homogenized in a 1.5-mL tube containing 1 mL of Trizol Reagent (Thermo Fisher Scientific). RNA isolation was followed in accordance with manufacturers instruction. RNA was resuspended in DEPC-treated RNase-free water (Thermo Fisher Scientific). TURBO DNA-free kit was used to remove residual DNA contamination according to manufacturers instruction (Thermo Fisher Scientific). 1 µg of total RNA was used for sequencing library preparation. PolyA-tailed RNAs were selected by NEBNext Poly(A) mRNA Magnetic Isolation Module (NEB), followed by the library prep using NEBNext Ultra RNA library Prep Kit for Illumina according to manufacturer's instruction (NEB). The libraries were checked and pooled as described above. Pooled denature libraries were sequenced on the illumina NextSeq 550 or Hiseq 2500 platforms with single end 100 bps.

## Quantitative PCR
### ChIP-qPCR
ChIP experiments were performed as described above. The same volume of input and IP DNA were used as template for quantitative PCR experiment. Analysis was performed using the QuantStudio 6 Flex real-time PCR system with SYBR selected master mix (Thermo Fisher Scientific). The percentage of input method was used for data analysis. Primers for target genes were listed in *Supplementary file 5*.

### RT-qPCR
RNA extractions were done as described above. 1 µg of total RNA was used for reverse transcription by random primers using SuperScript III First-strand synthesis system for RT-PCR (Thermo Fisher Scientific). Analysis was performed using the QuantStudio 6 Flex real-time PCR system with SYBR selected master mix (Thermo Fisher Scientific). The 2$^{-\Delta\Delta CT}$ method was used for quantification upon normalization to the RP49 gene as internal control. Primers for target genes were listed in *Supplementary file 5*.

## Metabolome analysis

### Metabolites extraction

Twenty heads for each replicates were collected into a microcentrifuge tube and immediately frozen into liquid nitrogen. For each experiment, 10 biological replicates were prepared. Fly heads were homogenized with 200 μl of $H_2O$ and 20 ceramic beads (0.1 mm) using the homogenizer (Precellys 24, Bertin Technologies). 800 μl of ACN:MeOH (1:1, v/v) was added for metabolite extraction. Vortex was for 30 s, followed by incubation in liquid nitrogen for 1 min. This freeze-thaw cycle was repeated three times. To precipitate protein, samples were incubated for 1 hr at −20℃, followed by 15 min centrifugation using 15,000 g at 4℃. The supernatant was removed and evaporated to dryness in a vacuum concentrator (Labconco, German). Dry extracts were then reconstituted in 100 μl of ACN:$H_2O$ (1:1, v/v), followed by 10 min sonication (50 Hz, 4℃) and 15 min centrifugation using 13,000 rpm at 4℃ to remove insoluble debris. Supernatants were transferred to HPLC glass vials and stored at −80℃ prior to LC/MS analysis.

### Lc-ms/MS

LC-MS was performed using a UHPLC system (1290 series, Agilent Technologies) coupled to a quadruple time-of-flight mass spectrometer (AB Sciex TripleTOF 6600). Waters ACQUITY UPLC BEH Amide columns (particle size, 1.7 μm; 100 mm (length) ×2.1 mm (i.d.)) were used. Mobile phases A = 25 mM ammonium acetate and 25 mM ammonium hydroxide in 100% water, and B = 100% acetonitrile, were used for both ESI positive and negative modes. And the linear gradient eluted from 95% B (0.0–1.0 min), 95% B to 65% B (1.0–14.0 min), 65% B to 40% B (14.0–16.0 min), 40% B (16.0–18.0 min), 40% B to 95% B (18.0–18.1 min), then stayed at 95% B for 4.9 min. The flow rate was 0.3 mL/min and the sample injection volume was 2 μL.

ESI source parameters on TripleTOF 6600 were set as followings: ion source gas 1 (GS1), 60 psi; ion source Gas 2 (GS2), 60 psi; curtain gas (CUR), 30 psi; temperature (TEM), 600℃; ion spray Voltage floating (ISVF), 5000 V or −4000 V, in positive or negative modes, respectively; declustering potential (DP), 60 V or −60 V in positive or negative modes, and collision energy for Product Ion, 30 eV or −30 eV, in positive or negative modes, respectively; collision energy spread (CES), 0 eV. LC-MS/MS data acquisition was operated under information-dependent acquisition (IDA) mode. The instrument was set to acquire over the m/z range 60–1200 Da for TOF MS scan and the m/z range 25–1200 for product ion scan. The accumulation time for TOF MS scan was set at 0.20 s/spectra and product ion scan at 0.05 s/spectra. The unit resolution was selected for precursor ion selection, and the collision energy (CE) was fixed at 30 V and −30 V for positive or negative modes, respectively. Collision energy spread (CES) was set as 0. IDA settings which were set as followings: charge state 1 to 1, intensity 100 cps, exclude isotopes within 4 Da, mass tolerance 10 ppm and maximum number of candidate ions 6. The 'exclude former target ions' was set as 4 s after two occurrences. In IDA Advanced tab, 'dynamic background subtract' was also chosen.

## Qualitative metabolic flux analysis using U-[$^{13}$C]-glucose as a tracer

### Metabolites extraction

Metabolites were extracted as described above for untargeted metabolomics.

### LC-MS

LC-MS analysis was performed using a UHPLC system (1290 series, Agilent Technologies) coupled to a quadruple time-of-flight mass spectrometer (6550 series, Agilent Technologies). Merck SeQuant ZIC-pHILIC column (particle size, 5 μm; 100 mm (length) ×2.1 mm (i.d.)) was used. Mobile phases A = 25 mM ammonium acetate and 25 mM ammonium hydroxide in 100% water, and B = 100% acetonitrile, were used for both ESI positive and negative modes. And the linear gradient eluted from 80% B (0.0–2.0 min, 0.2 mL/min), 80% B to 20% B (2.0–17.0 min, 0.2 mL/min), 20% B to 80% B (17.0–17.1 min, 0.2 mL/min), 80% B (17.1–22.1 min, 0.4 mL/min), 80% B to 80% B (22.1–22.2 min, 0.2 mL/min). The sample injection volume was 2 μL.

ESI source parameters on QTOF 6550 were set as followings: sheath gas temperature, 300℃; dry gas temperature, 250℃; sheath gas flow, 12 L/min; dry gas flow, 16 L/min; capillary voltage, 2500 V or −2500 V in positive or negative modes, respectively; nozzle voltage, 0 V; and nebulizer pressure,

20 psi. The MS1 data acquisition frequency was set as 4 Hz, and the TOF scan range was set as m/z 60–1200 Da.

## Data analysis

Data processing for qualitative metabolic flux analysis was performed using VistaFlux software suite from Agilent Technologies. First, Pathways to PCDL software (version B.07.00, Agilent Technologies) and PCDL Manager software (version B.07.00, Agilent Technologies) was used to build a metabolite library for metabolites in both glycolysis and citric acid cycle. Then, acquired LC-MS raw data files (.d) were loaded into Profinder (version B.08.00, Agilent Technologies) for the extraction of metabolite isotopologues using the constructed metabolite library. The ion abundance criterion for data extraction was set as 'use peak core area 20% of peak height'. The qualification parameters were set as: mass tolerance: $\pm15$ ppm $+2.00$ mDa; retention time tolerance :$\pm0.20$ min; anchor ion height threshold: 250 counts; sum of ion heights threshold: 1000 counts; correlation coefficient threshold: 0.5. The final results were exported into a csv file for the subsequent comparative and statistical analyses.

## Specific metabolites tests

Ten fly muscles were harvested and immediately frozen in liquid nitrogen. Tissues were homogenized using the homogenizer (Precellys 24, Bertin Technologies) with 300 µl ddH$_2$O or extraction buffer provided by the kit. For glucose, ATP, GSH and pyruvate assays, tissues were homogenized with ddH$_2$O, then frozen at $-80°$C for 30 min and thawed at room temperature, followed by 5 cycles of 30 s on and 30 s off sonication. For NADH and NADPH assays, tissues were homogenized with extraction buffer provided by the kit. Lysates were centrifuged using 10,000 $g$ at 4°C for 5 min and the supernatant was transferred to 10 kDa molecular weight cut-off spin filter (Sigma), followed by centrifugation using 10,000 $g$ at 4°C for 30 min for deproteinization. Assays were assembled according to the instruction of manufacturers (Glucose [Sigma, MAHK20], ATP [Promega, FF2000], GSH [Abcam, ab13881], pyruvate [Sigma, MAK071], NADH [Sigma, MAK037], NADPH [Sigma, MAK038]). Signals were detected by the EnSpire Multimode Plate Reader (PerKin Elmer).

## High-throughput sequencing data analysis

Sequencing reads were mapped to the reference genome dm6 or mm10, respectively, with Bowtie2-2.2.9 (*Langmead and Salzberg, 2012*) using default parameters. Samtools-1.3.1 (*Li et al., 2009*) was used for sam to bam format conversion. Peak regions were identified by homer-v4.8.3 (*Heinz et al., 2010*) function *findPeaks* with parameter '-style histone -F 2 -size 3000 -minDist 5000'. The overlapped peaks of replicate datasets were found using homer function *mergePeaks* using default parameter. Peak annotation was performed by homer function *annotatePeaks* with default parameter. The intersecting peak regions between different genotypes and aging stages were identified by BEDTools-2.26.0 (*Quinlan and Hall, 2010*) using default parameter.

For quantitative comparison, the dm6 mapped reads were normalized to a scale factor using deeptools-2.2.4 (*Ramírez et al., 2014*) function *bamCoverage* with 10 bp bin size. The scale factor for each sample was calculated as reported in *Orlando et al., 2014* with a small modification by using the percentage of mapped spike-in mouse genome (genome build mm10) in input from which we derived the scale factor, *a*, as follows:

Let us define *a* as the normalized factor, *b* as the reference signal from the mouse cells, *c* as the total number of reads (in millions) from a sample aligned to mm10 genome, and *d* as the percentage of mapped spike-in mouse genome (genome build mm10) in input.

Based on our experimental design, the fraction of the spike-in mouse genome over the total fly genome is a constant. Mathematically, *b* can be expressed as follows:

$$b = a \times c/d \qquad (1)$$

The epigenome of the reference sample does not vary appreciably, such that the reference signal from mouse cell should be the same across all sample being examined. Taking *equation (1)* and the fact that b is constant across all the samples, it is easy to show that:

$$a \propto d/c \qquad (2)$$

Without loss of generality, we can set the value of b to one and the scale factor $a$ can be estimated by:

$$a = d/c$$

The ChIP intensity for the gene bodies (transcriptional start site to transcriptional termination sites as annotated in dm6) of all protein-coding genes or peak region was calculated using the bwtool (*Pohl and Beato, 2014*) function *summary* with default parameters. The input bigwig file, normalized by a scale factor, was generated by *bamCoverage* function. Bootstrapping was used for calculating the mean reference-adjusted ChIP intensity of peak genes or inter-peak genes (10000 draws with replacement of n = 500) and the confidence intervals. Violin plots represent the bootstrapped mean H3K27me3 level of inter-peak genes and peak genes at different ages. The scatter plot and violin plot was generated by R package ggplot2.

For visualization of the selected genome regions, we used IGV-2.3.31 (*Robinson et al., 2011*) with the bigwig files generated by *bamCoverage*. Circos plot was generated by J-circos-V1 (*An et al., 2015*) with the bigwig files. Cistrome correlation tools (*Liu et al., 2011*) were used to generate the heatmap with hierarchical clustering (*hclust* function in R) according to pair-wise correlation coefficients. Pair-wise Pearson correlations of genome-wide H3K27me3 levels were calculated at windows of 1Kbps on the genome scale using multiple bigwig file. The deeptools function *bamCompare* was used to generated bigwig file based on ChIP and input bam files that are compared to each other with ratio while being simultaneously normalized using RPKM.

## PolyA-selected RNA-seq

Sequencing reads were mapped to the reference genome dm6 with STAR2.3.0e (*Dobin et al., 2013*) by default parameter. The read counts for each gene were calculated by HTSeq-0.5.4e (*Anders et al., 2015*) htseq-count with parameters '-m intersection-strict -s no' with STAR generated SAM files. The count files were used as input to R package DESeq (*Anders and Huber, 2010*) for normalization and the differential expression genes were differential expression genes were computed based on normalized count from three biological replicates (p<0.05). The venn diagrams of overlapped differential genes were generated by VENNY-2.1 (http://bioinfogp.cnb.csic.es/tools/venny/index.html). GO term analysis was performed by David (*Huang et al., 2009a*, *Huang et al., 2009b*) with default parameters. WGCNA (1.51) was used for network analysis of co-regulated genes across PRC2 mutant samples with parameters "Soft thresholding power $\beta$ = 8; Scale-free topology fit index $R^2$ = 0.897). For selected module, Cytoscape (3.5.1) was used for visualizing interaction network.

## Metabolome

The raw MS data (.wiff) were converted to mzML files using ProteoWizard MSConvert (version 3.0.6526) and processed using XCMS (*Smith et al., 2006*) for feature detection, retention time correction and alignment. Part of XCMS processing parameters were optimized and set as followings: mass accuracy in peak detection = 25 ppm; peak width c = (5, 20); snthresh = 6; bw = 10; minfrac = 0.5. R package CAMERA (*Kuhl et al., 2012*) was used for peak annotation after XCMS data processing. A customized R script was used to preprocess and extract the MS/MS data for database match with in-house library. The resulting raw data was then preprocessed for data normalization using SVR algorithm (*Shen et al., 2016*). Pathway enrichment analysis was executed using MetaboAnalyst 3.0 (*Xia et al., 2015*). In MetaboAnalyst 3.0, metabolic pathways for *Drosophila melanogaster* were selected. The pathway analysis algorithms were hypergeometric test for over representation analysis and relative-betweeness centrality for pathway topology analysis. For normalized intensity calculation, the preprocessed metabolomic data was further scaled (see below).

$$\text{Normalized intensity} = \frac{x}{\sigma}$$

$\sigma$ is the standard deviation of the population.

Specifically, in each sample group, the pathway intensity was determined by calculating the sum of detected metabolites' normalized intensities in each metabolic pathway.

## Data and software availability

### Data resources

The raw data files of sequencing experiments have been deposited in the NCBI Gene Expression Omnibus, as well as the normalized read density profiles of ChIP-seq and differential expression results from DESeq of RNA-seq reported in this paper. The accession number is GEO:GSE96654.

### Quantification and statistical analysis

All statistical details of experiments are included in figure legends. Sample numbers and experiment replicates are stated in figure legends.

## Acknowledgements

We thank Drs. Jilong Liu, Yun Zhao, Meng-Qiu Dong, Cong Liu, Chao Tong, Xing Guo, Lanfeng Wang, and Ye Tian for advice on the manuscript; Wei Wu and the Core Facility of *Drosophila* Resource and Technology, Shanghai Institute of Biological Sciences, Chinese Academy of Sciences, Shanghai, China for fly microinjections. Prof. Junying Yuan provided considerable support, advice on the experiments, and critical suggestions on the manuscript. NL and ZZ are Junior Scholar of the 1000 Plan of China. YZ is supported by 100 Talents Program of the Chinese Academy of Sciences. This work was supported by grants from the National Program on Key Basic Research Project of China to NL and YZ (2016YFA0501900); the National Natural Science Foundation of China to NL (31371326), YZ (31671428, 31500665, 31530041), ZZ (21575151), and FL (81770143); a National Institute of Health grant (GM120033) to ZL; a National Science Foundation grant (DMS-1263932) to ZL; a Cancer Prevention Research Institute of Texas grant (RP170387) to ZL; the Robert A and Renée E Belfer Family Foundation to ZL, and the Chao Family Foundation to ZL.

## Additional information

### Competing interests

Rui Liu: is affiliated with Singlera Genomics, a company providing customized next generation sequencing services. The author has no financial interests to declare. The other authors declare that no competing interests exist.

### Funding

| Funder | Grant reference number | Author |
| --- | --- | --- |
| National Natural Science Foundation of China | 81770143 | Feng Liu |
| National Institutes of Health | GM120033 | Zhandong Liu |
| National Science Foundation | DMS-1263932 | Zhandong Liu |
| Cancer Prevention and Research Institute of Texas | RP170387 | Zhandong Liu |
| Robert A and Renée E Belfer Family Foundation | | Zhandong Liu |
| Chao Family Foundation | | Zhandong Liu |
| National Natural Science Foundation of China | 31671428 | Yaoyang Zhang |
| National Natural Science Foundation of China | 31500665 | Yaoyang Zhang |
| National Natural Science Foundation of China | 31530041 | Yaoyang Zhang |
| National Program on Key Research Projects of China | 2016YFA0501900 | Yaoyang Zhang Nan Liu |

| | | |
|---|---|---|
| 100 Talents Program of the Chinese Academy of Sciences | | Yaoyang Zhang |
| National Natural Science Foundation of China | 21575151 | Zheng-Jiang Zhu |
| National Natural Science Foundation of China | 31371326 | Nan Liu |

The funders had no role in study design, data collection and interpretation, or the decision to submit the work for publication.

## Author contributions
Zaijun Ma, Investigation, Writing—original draft; Hui Wang, Investigation, Methodology, Writing—original draft; Yuping Cai, Investigation, Methodology; Han Wang, Kongyan Niu, Xiaofen Wu, Huanhuan Ma, Yun Yang, Wenhua Tong, Investigation; Feng Liu, Zhandong Liu, Funding acquisition, Investigation; Yaoyang Zhang, Funding acquisition, Investigation, Methodology; Rui Liu, Methodology; Zheng-Jiang Zhu, Funding acquisition, Methodology; Nan Liu, Conceptualization, Funding acquisition, Visualization, Writing—original draft, Project administration, Writing—review and editing

## Author ORCIDs
Hui Wang (ID) http://orcid.org/0000-0002-3522-0164
Zheng-Jiang Zhu (ID) http://orcid.org/0000-0002-3272-3567
Nan Liu (ID) http://orcid.org/0000-0002-7384-0794

## Decision letter and Author response
Decision letter https://doi.org/10.7554/eLife.35368.029
Author response https://doi.org/10.7554/eLife.35368.030

## Additional files

### Supplementary files
• Supplementary file 1. Summary of H3K27me3 peak regions. Summary of ChIP-seq experiments. Two criteria for peak calling: IP/input $\geq$ 2, signals spanning $\geq$3 kb. In total, 222 peak regions of H3K27me3 were reproducibly identified by four biological replicates at 3d. For each peak regions, information pertaining to chromosomal location and genes therein contained were given. ChIP-seq was using muscle tissues of 3d=old animals. Genotype: 5905.
DOI: https://doi.org/10.7554/eLife.35368.020

• Supplementary file 2. Summary of CRISPR/Cas9-led gene mutagenesis and lifespan. For each gene, two sgRNAs with indicated sequences, detailed deletion sites confirmed by Sanger sequencing, and PCR validation were shown. To name new CRISPR mutant, a superscript amended to the gene contained a letter c denoting CRISPR/Cas9-directed mutagenesis followed by the size of genomic deletion. Mutants were backcrossed (at least five generations) into 5905 (Flybase ID FBst0005905, w[1118]) to assure that phenotypes were not associated with any variation in background. Lifespan curves, together with 50% (median lifespan) and 10% survival data were listed. (for lifespan assay: 25°C; n $\geq$ 140 per genotype for curve; log-rank test).
DOI: https://doi.org/10.7554/eLife.35368.021

• Supplementary file 3. Summary of transcriptomic analysis of PRC2 longevity mutants. RNA-seq summary. This table summarizes five categories of genes and their transcriptional changes in long-lived PRC2 mutants of indicated genotype: (1) commonly upregulated and (2) downregulated genes in all PRC2 long-lived mutants; genes known to promote longevity upon upregulation, including (3) insulin pathway genes, (4) TOR signaling pathway genes, and (5) genes of miscellaneous pathways. RNA-seq was from muscle and head tissues of 30d-old male flies.
DOI: https://doi.org/10.7554/eLife.35368.022

• Supplementary file 4. Summary of transcriptomic analysis of genes in glycolysis, citric acid cycle, pentose phosphorylation pathway, and oxidative phosphorylation. This table summarizes metabolic

genes and their transcriptional changes in long-lived PRC2 mutants of indicated genotype. RNA-seq was from head and muscle tissues of 3d- and 30d-old male flies.

DOI: https://doi.org/10.7554/eLife.35368.023

• Supplementary file 5. Supplemental Materials and methods: Primer sequences used for the study.

DOI: https://doi.org/10.7554/eLife.35368.024

• Transparent reporting form

DOI: https://doi.org/10.7554/eLife.35368.025

## Data availability

The raw data files of sequencing experiments have been deposited in the NCBI Gene Expression Omnibus, as well as the normalized read density profiles of ChIP-seq and differential expression results from DESeq of RNA-seq reported in this paper. The accession number is GEO: GSE96654.

The following dataset was generated:

| Author(s) | Year | Dataset title | Dataset URL | Database, license, and accessibility information |
|---|---|---|---|---|
| Liu N | 2018 | Raw data of sequencing experiments, normalized read density profiles of ChIP-seq, and the differential expression results from DESeq of RNA-seq | http://www.ncbi.nlm.nih.gov/geo/query/acc.cgi?acc=GSE96654 | Publicly available at the NCBI Gene Expression Omnibus (accession no: GSE96654) |

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
