## [Decision Letter]

Thank you for submitting your article "Epigenetic Drift of H3K27me3 in Aging Links Glycolysis to Healthy Longevity" for consideration by *eLife*. Your article has been favorably evaluated by Kevin Struhl (Senior Editor) and three reviewers, one of whom, Matt Kaeberlein (Reviewer #1), is a member of our Board of Reviewing Editors. The following individual involved in review of your submission has agreed to reveal their identity: Adrienne Wang (Reviewer #3).

The reviewers have discussed the reviews with one another and the Reviewing Editor has drafted this decision to help you prepare a revised submission.

Summary:

Overall, the referees appreciated the novelty and importance of the work and the provocative new hypotheses suggested. Several major concerns were noted, however, that should be addressed before publication. Many of these concerns involve more thorough integration of these studies within the existing literature and a discussion of how the results fit, or in some cases run counter to, prevailing models for aging. In particular, this study is limited in that it may be unique to *Drosophila*. While the reviewers found the data to be credible, the proposed model does contradict data in other organisms regarding the importance of glycolysis in aging (large body of CR literature, work from Ristow in worms and mice https://www.nature.com/articles/ncomms4563) and is directly contradicting the current thinking about NAD/NADH ratio and aging. The authors at least need to acknowledge these contradictions and the possibility that this is a private mechanism of aging in *Drosophila* that will not be conserved in other organisms. Unless they have data to suggest that this is a conserved mechanism of aging, in which case it is strongly recommended that they add it to the paper.

There is also a striking absence of discussion of prior literature related to epigenetic marks modulating aging and lifespan in yeast and worms. The studies from Shelley Berger's lab showing that histone acetylation impact aging in yeast should be discussed, as should the work from Anne Brunet and others in *C. elegans* and in stem cells, some of which specifically deals with H3K27me3. The authors need to put their work into context with the prior literature rather than ignoring it. To claim that this study shows that H3K27me3 is a "new mechanism that drives the progression of aging" would seem an overstatement in light of the prior literature in this area. As another example of missing literature, the authors completely ignore a study by the Rabinovitch group in which quantitative proteomics was used to characterize age-related changes in the heart proteome and in which glycolytic enzymes were shown to increase with age, while calorie restriction was shown to decrease the abundance of these enzymes, promoting health of this tissue (Dai et al., 2014).

The referees also felt that in many cases, the data were over-interpreted. For example, the subsection “Long-lived PRC2 Mutants Diminish the Epigenetic Drift of H3K27me3 During Aging”, ends with 'Combined, our study implicates that the life-extension effect of PRC2-mutation likely arises from its ability to mitigate the drifting of H3K27me3 during aging.' At this point in the paper, it has only been shown that PRC2 mutants live longer and that they have reduced H3K27me3 levels. A functional or causal connection between these phenotypes has not been established, and the conclusion is thus not valid. There are many other instances of such over-interpretation, and the authors need to edit the text significantly to more conservatively and accurately interpret the results.

In addition, there are a few experimental studies, additional statistical analyses, and additional textual changes that were considered to be essential for further consideration at *eLife*.

Essential revisions:

1) There is the lack of detail on the collection, processing, and statistical analysis of the large datasets. It is therefore difficult to evaluate how rigorous the analysis of these data was. Raw datasets (for example FPKM tables for the RNAseq results, as well as cohort data for the lifespan studies – for which statistical analysis is wholly missing) should be made available to evaluate the data. Further consultation with a statistician before resubmission is highly recommended.

2) A majority of the results are based on muscle samples, but the authors occasionally use head-specific samples throughout the manuscript. Though both tissues have high energy demand, their regulation of metabolism is very different. This is something that should be acknowledged in the manuscript. In particular, since epigenetic marks are very cell-type dependent, a clearer characterization and description of epigenetic changes specifically in muscle and brain tissue is warranted. Based on the Materials and methods section, the data was acquired from muscle, but later RNAseq studies also include the brain. Tissue-specificity needs to be clearly described and if transcriptome or metabolomic data from brain is used, then the ChIPseq data should be obtained from brain as well.

3) The metabolomics data raises important unresolved questions that could be addressed. The authors measured a total of 151 metabolites. This likely represents far less than the total metabolome found in *Drosophila*. The size of their dataset is critical in drawing inference about the key pathways that are associated with lifespan. The authors do look specifically for changes in the TCA cycle, but might there be other equally predictive pathways associated with metabolism that are not represented here given the small number of metabolites measured overall? What happens to TCA intermediates? (ATP levels are down, suggesting glycolysis is the main source of ATP, but this isn't actually measured. What happens to oxphos activity/efficiency? What is happening to the PPP pathway?

4) How does the global loss of PRC2 affect the developmental establishment of H3K27me3 patterns (i.e. do certain marks never get established in the first place? It is mentioned that 'global occupancy of H3K27me3 between wt and age-matched PRC2 mutants was highly preserved', but the correlation coefficient is only 0.77, suggest that there are significant changes in certain loci that are being ignored). A more acute depletion of PRC2 in the adult, for example by RNAi would be useful to analyze its effects specifically on aging-related pattern changes. It would also be important to acutely deplete PRC1 to assess the specificity of the PRC2 effects. And to perturb both PRC1 and PRC2 to establish whether the establishment and/or maintenance of epigenetic marks is affected by age. It is shown that PRC1 mutants have extended lifespan but no changes in H3K27me3, and this discrepancy with PRC2 mutants is ignored – if PRC1 mutants are also long-lived, it seems likely that the longevity effects are not caused by changes in H3K27me3.

5) As this paper is reliant upon accurate metabolic analysis, and the levels of protein and sugar in *Drosophila* food have documented effects on lifespan and metabolism, the authors should specify the "standard" food their flies are reared on.

6) Given that the authors are studying changes that occur with aging, age is an important factor for the interpretation of most of the figures. Throughout the manuscript, there are many places where age of samples used is not explicitly stated, or is unclear – for example, the samples used for the transcriptomic analysis.

7) The authors show that down-regulation of these PRC2-mediated glycolytic genes blocks the lifespan extension seen in PRC2 mutants, thus establishing that the mechanism of lifespan extension by PRC2 is at least partially reliant upon glycolysis. However, the extent to which downregulation of these genes blocks the PRC2-mediated lifespan extension is unclear from the text and without referring to supplementary figures. As there was only partial reduction of lifespan by knocking down glycolytic genes in comparison to WT, it indicates that other mechanisms might be in play mediating lifespan in this model. This should be explicitly addressed.

8) The authors should rearrange the figures such that the actual data are shown in Figure 2 and the summary table is moved to supplemental.

9) The observations related to NADH/NAD ratio run counter to a growing body of literature suggesting that NADH/NAD ratio increases during aging and that restoring this imbalance (for example through NAD precursors) can increase lifespan and/or healthspan. The authors should at least comment on this apparent discrepancy and offer some speculation about why they are getting an opposite result.

10) The statement "drives the progression of aging" is perhaps overly reaching. It would be better to say that the data presented in this study are consistent with the model that epigenetic drift of H3K27me3 is one of the molecular mechanisms that contributes to at least a subset of age-associated phenotypes and limits lifespan in fruit flies.

11) Typo in the subsection “A CRISPR/Cas9 Deficiency Screen Identifies the Role of PRCs in Aging”: Should read "Given that the above study indicated…"

12) Figure 1C and D: it is unclear why red highlighted area extends below diagonal line, if the authors claim that signals in aged animals are higher than in younger? (same goes for some of the other scatter plots…)

13) Figure 4C: what is the age of analyzed animals?

14) Locomotion: usually longer distances than 2cm are recorded (e.g. PMID:25850672 uses 8cm in 20 seconds). Any specific reason for short distance?

---

## [Author Response]

Summary:Overall, the referees appreciated the novelty and importance of the work and the provocative new hypotheses suggested. Several major concerns were noted, however, that should be addressed before publication. Many of these concerns involve more thorough integration of these studies within the existing literature and a discussion of how the results fit, or in some cases run counter to, prevailing models for aging. In particular, this study is limited in that it may be unique to Drosophila. While the reviewers found the data to be credible, the proposed model does contradict data in other organisms regarding the importance of glycolysis in aging (large body of CR literature, work from Ristow in worms and mice https://www.nature.com/articles/ncomms4563) and is directly contradicting the current thinking about NAD/NADH ratio and aging. The authors at least need to acknowledge these contradictions and the possibility that this is a private mechanism of aging in Drosophila that will not be conserved in other organisms. Unless they have data to suggest that this is a conserved mechanism of aging, in which case it is strongly recommended that they add it to the paper.There is also a striking absence of discussion of prior literature related to epigenetic marks modulating aging and lifespan in yeast and worms. The studies from Shelley Berger's lab showing that histone acetylation impact aging in yeast should be discussed, as should the work from Anne Brunet and others in C. elegans and in stem cells, some of which specifically deals with H3K27me3. The authors need to put their work into context with the prior literature rather than ignoring it. To claim that this study shows that H3K27me3 is a "new mechanism that drives the progression of aging" would seem an overstatement in light of the prior literature in this area. As another example of missing literature, the authors completely ignore a study by the Rabinovitch group in which quantitative proteomics was used to characterize age-related changes in the heart proteome and in which glycolytic enzymes were shown to increase with age, while calorie restriction was shown to decrease the abundance of these enzymes, promoting health of this tissue (Dai et al., 2014).The referees also felt that in many cases, the data were over-interpreted. For example, the subsection “Long-lived PRC2 Mutants Diminish the Epigenetic Drift of H3K27me3 During Aging”, ends with 'Combined, our study implicates that the life-extension effect of PRC2-mutation likely arises from its ability to mitigate the drifting of H3K27me3 during aging.' At this point in the paper, it has only been shown that PRC2 mutants live longer and that they have reduced H3K27me3 levels. A functional or causal connection between these phenotypes has not been established, and the conclusion is thus not valid. There are many other instances of such over-interpretation, and the authors need to edit the text significantly to more conservatively and accurately interpret the results.

We thank the reviewer for highlighting the significance of the work and the impact to the field. Given the scope of our data, we have now changed the title of the manuscript into: Epigenetic Drift of H3K27me3 in Aging Links Glycolysis to Healthy Longevity in *Drosophila*. We appreciate the issue raised by the reviewer. In the revised manuscript, we have substantially re-written the background and Discussion to better fit into and further advance our current understanding of aging.

In Abstract: we have edited as follows: “we propose that epigenetic drift of H3K27me3 is one of the molecular mechanisms that contribute to aging”

In Introduction: we have re-written as follows:

“Previous studies have established that changes in histone markings, such as acetylation on histone H4 at lysine 16 (H4K16ac) in *S. cerevisiae* (Dang et al., 2009), trimethylation on histone H3 at lysine 4 (H3K4me3) in *C. elegans* (Han et al., 2017), and trimethylation on histone H3 at lysine 36 (H3K36me3) in *S. cerevisiae, C. elegans* and *D. melanogaster* substantially impact adult lifespan (Pu et al., 2015; Sen et al., 2015).”

“Though mutation in the H3K27me3 demethylase couples an increase of H3K27me3 with longevity in *C. elegans* (Jin et al., 2011; Maures et al., 2011), reducing H3K27me3 by heterozygous mutation in the H3K27me3 methyltransferase results in lifespan extension in both *C. elegans* and *D. melanogaster* (Ni et al., 2012; Siebold et al., 2010). Hence, it remains unclear whether the life-benefit phenotype is due to manipulation of specific genes or modulatory effect of H3K27me3.”

“Our data presented in this study are consistent with the model that epigenetic drift of H3K27me3 is one of the molecular mechanisms that contribute to at least a subset of age-associated phenotypes and limit lifespan in *Drosophila*.”

In Discussion:we have re-written as follows:

“Quantitative proteomic analysis in mouse heart shows that aging correlates with an increased abundance of proteins in glycolysis, and that caloric restriction and rapamycin treatment known to extend longevity can reverse this trend (Dai et al., 2014). D-Glucosamine supplementation that impairs glucose metabolism can extend lifespan in *C. elegans* and mouse (Weimer et al., 2014).”

“It is important to note that since our study is based on *Drosophila* somatic tissues that are primarily post-mitotic cells in adult animals, it is possible that the effects are more dominant in the fly model than others.”

“Aging is a complex process that can be regulated by a network of multiple mechanisms. It has been well-established that enhancing NAD^+^ biogenesis promotes healthy lifespan (Anderson et al., 2002; Balan et al., 2008; Mills et al., 2016). […] Intriguingly, while declining intracellular NAD^+^ and thusincreased NADH/NAD^+^ correlate with aging (Zhu et al., 2015), our experiments in *Drosophila* demonstrate that increased ratios of GSH/(GSH^+^GSSG) and NADH/NAD^+^ due to enhanced glycolytic activities may provide a simple but effective way to retard aging.”

Essential revisions:1) There is the lack of detail on the collection, processing, and statistical analysis of the large datasets. It is therefore difficult to evaluate how rigorous the analysis of these data was. Raw datasets (for example FPKM tables for the RNAseq results, as well as cohort data for the lifespan studies – for which statistical analysis is wholly missing) should be made available to evaluate the data. Further consultation with a statistician before resubmission is highly recommended.

We apologize for lack of clarity in presentation of our work. In the revised manuscript, we have added statistic method next to the figure panels and the figure legends; the Materials and methods section also includes more details, including data collection, processing, and statistical analysis.

Both raw and processed data files of ChIP-seq and RNA-seq experiments have been deposited in the NCBI Gene Expression Omnibus (accession number: GEO:GSE96654, the following secure token has been created to allow review of record GSE96654 while it remains in private status: adotmsqctzerxol). A detailed lifespan data, including fly number, lifespan curve, median and 10% survival, and statistics, is provided in Supplementary file 2.

Dr. Zhandong Liu, a co-author of the manuscript, is an expert in biostatistician and now an assistant professor at the Baylor College of Medicine, Houston, USA, who helped to oversee the procedures of data processing and analysis.

2) A majority of the results are based on muscle samples, but the authors occasionally use head-specific samples throughout the manuscript. Though both tissues have high energy demand, their regulation of metabolism is very different. This is something that should be acknowledged in the manuscript. In particular, since epigenetic marks are very cell-type dependent, a clearer characterization and description of epigenetic changes specifically in muscle and brain tissue is warranted. Based on the Materials and methods section, the data was acquired from muscle, but later RNAseq studies also include the brain. Tissue-specificity needs to be clearly described and if transcriptome or metabolomic data from brain is used, then the ChIPseq data should be obtained from brain as well.

For each experiment, we have carefully labeled tissue types used next to the figure panel, in the text and figure legends. Moreover, we have profiled H3K27me3 using dissected heads of *Pcl*^c421^*Su(z)12*^c253^ mutants and their age-matched WT controls, as shown in revised Figure 4 and Figure 4—figure supplement 1. Similar to what we have noted in muscle, our data shows that H3K27me3 in heads has age-associated increase and drift during normal aging in WT animals, and that this shift is partially diminished by PRC2-mutation.

To find transcriptional change of particular genes that might play a dominant role in PRC2-dependent life-extension, we compared RNA-seq datasets derived from PRC2 mutants in both head and muscle tissues. This analysis led us to narrow down to two glycolytic genes, *Tpi* and *Pgi*, whose expressions were upregulated in both tissue types in all long-lived PRC2 mutants. For untargeted metabolomics and metabolic flux experiments, we used dissected heads to avoid metabolic features carried over from the food residues in the trachea or the gut. For measurement of specific metabolites, we used dissected muscles. Our data consistently supports our conclusion that glycolysis is modulated with age and in PRC2-deficiency.

3) The metabolomics data raises important unresolved questions that could be addressed. The authors measured a total of 151 metabolites. This likely represents far less than the total metabolome found in Drosophila. The size of their dataset is critical in drawing inference about the key pathways that are associated with lifespan. The authors do look specifically for changes in the TCA cycle, but might there be other equally predictive pathways associated with metabolism that are not represented here given the small number of metabolites measured overall? What happens to TCA intermediates? (ATP levels are down, suggesting glycolysis is the main source of ATP, but this isn't actually measured. What happens to oxphos activity/efficiency? What is happening to the PPP pathway?

For the metabolomics data set, about 20,000 metabolic features were detected in total, including adducts and isotopes corresponding to the unique metabolite ID. Approximately 30% of metabolic features with MS/MS spectrum were obtained using data-dependent acquisition (DDA) based untargeted metabolomics approach. Among all putatively characterized metabolites, 151 metabolites matched against our standard library were accurately identified with the highest confidence level (Level 1) according to Metabolomics Standards Initiative (MSI). The standard library for spectral match consists of 830 metabolites covering a majority of metabolism pathways in KEGG database. Therefore, the metabolites identified in *Drosophila* samples are unbiased benefiting from the relative comprehensive library. It is important to note that, in addition to unbiased metabolic analysis, we also have metabolic flux experiment (Figure 6 and Figure 6—figure supplement 2) in which we observed no consistent change in the metabolites of the TCA cycle with age and in PRC2-deficiency, which was in line with constant levels of metabolites as measured by untargeted metabolomics as shown in Figure 6—figure supplement 1B. RNA-seq experiments of individual long-lived PRC2 mutants found no consistent changes in the expression of genes mediating pentose phosphate pathway (PPP), citric acid cycle, and oxidative phosphorylation (Figure 5—figure supplement 1C, Supplementary file 4). These data together led us to conclude that expression and metabolism of the glycolytic pathway is affected with age and in PRC2-deficiency.

4) How does the global loss of PRC2 affect the developmental establishment of H3K27me3 patterns (i.e. do certain marks never get established in the first place? It is mentioned that 'global occupancy of H3K27me3 between wt and age-matched PRC2 mutants was highly preserved', but the correlation coefficient is only 0.77, suggest that there are significant changes in certain loci that are being ignored).

We would like to clarify that our data was based on studying heterozygous mutation of PRC2 genes, in which H3K27me3 modification, despite at a reduced level, remained operative in mutant animals. In Figure 3A, Circos plot of the H3K27me3 epigenome illustrates peak profiles that are highly preserved with age and in PRC2 mutants.

As noted by Liu et al. (Liu et al., 2011) and a web-based application called Cistrome (http://cistrome.org/ap/root) therein described, the correlation coefficient between +0.7 and +1.0 can be interpreted as strong positive correlation. Nevertheless, we agree with reviewers that we cannot fully rule out the possibility that unknown H3K27me3 targets might be affected by PRC2-deficiency. Thus we have modified the sentence to read as follows: “Interestingly, global occupancy of H3K27me3 between WT and age-matched PRC2 mutants was preserved, as demonstrated by the comparable patterns of peak profiles and genes known to be H3K27me3-modified (pair-wise Pearson correlation coefficients ≥0.77) (Figure 3A, Figure 3—figure supplement 1B and 1C), though we cannot rule out the possibility that certain H3K27me3 targets might be altered.”.

A more acute depletion of PRC2 in the adult, for example by RNAi would be useful to analyze its effects specifically on aging-related pattern changes. It would also be important to acutely deplete PRC1 to assess the specificity of the PRC2 effects. And to perturb both PRC1 and PRC2 to establish whether the establishment and/or maintenance of epigenetic marks is affected by age.

To knockdown PRC1/2 factors specifically in the adult, we have explored two methods. First, we fed flies with available chemical compounds targeting mammalian PRC2 genes (compound ID: GSK343, GSK503, GSK126, EPZ6438, EPZ5687, UNC1999, and EI1 from Selleck Chemicals, USA). Unfortunately, none of these compounds showed inhibitory effect against *Drosophila* PRC2 counterpart, as evidenced by their inability to decrease H3K27me3 levels. Alternatively, we combined adult-onset Gal4 driver (*geneswitch*-GAL4 promoter that can be induced by feeding adult flies with RU486) with pUAST-transgene expressing double stranded RNA against PRC1/2 genes. One RNAi line that we have carefully tested was THU0754 (Tsinghua University, China) against *Su(z)12* gene. Induction of this RNAi line caused 20% decrease of *Su(z)12* mRNAas compared to *geneswitch*-GAL4 control.However, for the lifespan experiment, we surprisingly noted that *geneswitch*-GAL4 flies alone had a dramatically shortened lifespan, to about 65% that of WT animals. Induction of *Su(z)12* RNAi had a modest 10% lifespan extension than *geneswitch*-GAL4 control animals. Though this experiment may seem supportive, we hesitated to further interpret this result because *geneswitch*-GAL4 alone and perhaps RU486 feeding have already induced strong, confounding aging phenotypes. We feel that there is something quite interesting about the adult-specific deletion of PRC1/2, although to resolve this is beyond the scope of the current studies. It is worth to note, however, that fly behaviors and lifespan phenotypes are particularly sensitive to genetic background. We highlight that *all* fly lines used in our studies have been either isogenized or directly generated in the control background 5905 (Flybase ID FBst0005905, *w1118*). These include altogether 29 new site-specific gene mutations as described in Supplementary file 2.

It is shown that PRC1 mutants have extended lifespan but no changes in H3K27me3, and this discrepancy with PRC2 mutants is ignored – if PRC1 mutants are also long-lived, it seems likely that the longevity effects are not caused by changes in H3K27me3.

While PRC2 complex is required for H3K27me3 modification, PRC1 complex is generally involved in maintenance of the silencing status. Our data that PRC1-deficiency has no effect in H3K27me3 levels is consistent with a prior study by Tie et al., where they observed no detectable change in H3K27me3 upon knockdown of *Polycomb* gene of the PRC1 complex (Tie et al., 2016). As a result, our CRISPR/Cas9 mutagenesis screen identifies longevity genes functionally related to H3K27me3, including factors of PRC2 in establishing H3K27me3 marks and PRC1 in maintenance of H3K27me3-led repressive function.

5) As this paper is reliant upon accurate metabolic analysis, and the levels of protein and sugar in Drosophila food have documented effects on lifespan and metabolism, the authors should specify the "standard" food their flies are reared on.

We apologize for this. We have detailed the food receipt in the method. Briefly, standard *Drosophila* food contains sucrose (36g/L), maltose (38g/L), yeast (22.5g/L), agar (5.4g/L), maizena (60g/L), soybean flour (8.25g/L), sodium benzoate (0.9g/L), methyl-p-hydroxybenzoate: (0.225g/L), propionic acid (6.18ml/L), and ddH_2_O to make up 1L of the food.

6) Given that the authors are studying changes that occur with aging, age is an important factor for the interpretation of most of the figures. Throughout the manuscript, there are many places where age of samples used is not explicitly stated, or is unclear – for example, the samples used for the transcriptomic analysis.

We apologize for this. For each experiment, we have now carefully labeled age of the samples used next to the figure panel, in the main text and figure legends as appropriate.

7) The authors show that down-regulation of these PRC2-mediated glycolytic genes blocks the lifespan extension seen in PRC2 mutants, thus establishing that the mechanism of lifespan extension by PRC2 is at least partially reliant upon glycolysis. However, the extent to which downregulation of these genes blocks the PRC2-mediated lifespan extension is unclear from the text and without referring to supplementary figures. As there was only partial reduction of lifespan by knocking down glycolytic genes in comparison to WT, it indicates that other mechanisms might be in play mediating lifespan in this model. This should be explicitly addressed.

We apologize for this, and we have added RT-qPCR data in Figure 8—figure supplement 1E and 1J, for *Tpi*^c511^ and *Pgi*^c392^, respectively. We would like to clarify that since homozygous *Tpi*^c511^ or *Pgi*^c392^ had a pre-adult lethality, we could only combine *Tpi*^c511^ heterozygote or *Pgi*^c392^ heterozygote with *Pcl*^c421^*Su(z)12*^c253^, another trans-heterozygote double mutants, and evaluated the adult phenotypes by using the triple mutants. This genetic experiment, though showing partial rescue effects, supports our conclusion that perturbing glycolysis by genetic mutation diminishes longevity traits in PRC2 mutants. However, we agree with reviewers that we cannot fully rule out the possibility that other mechanisms might be in play in mediating lifespan in this model. Thus we have modified the sentence to read as follows: “These data implicate that perturbing glycolysis by genetic mutation diminishes longevity traits in PRC2 Mutants. Given the effect is partial, we cannot rule out the possibility that other mechanisms might be involved.”

8) The authors should rearrange the figures such that the actual data are shown in Figure 2 and the summary table is moved to supplemental.

We have fixed it.

9) The observations related to NADH/NAD ratio run counter to a growing body of literature suggesting that NADH/NAD ratio increases during aging and that restoring this imbalance (for example through NAD precursors) can increase lifespan and/or healthspan. The authors should at least comment on this apparent discrepancy and offer some speculation about why they are getting an opposite result.

The question being raised here is an important one and we have rewritten the Discussion to clarify this point, as follows: “Aging is a complex process that can be regulated by a network of multiple mechanisms. […] Intriguingly, while declining intracellular NAD^+^ and thusincreased NADH/NAD^+^ correlate with aging (Zhu et al., 2015), our experiments in *Drosophila* demonstrate that increased ratios of GSH/(GSH^+^GSSG) and NADH/NAD^+^ due to enhanced glycolytic activities may provide a simple but effective way to retard aging.”

10) The statement "drives the progression of aging" is perhaps overly reaching. It would be better to say that the data presented in this study are consistent with the model that epigenetic drift of H3K27me3 is one of the molecular mechanisms that contributes to at least a subset of age-associated phenotypes and limits lifespan in fruit flies.

We have re-written the text.

11) Typo in the subsection “A CRISPR/Cas9 Deficiency Screen Identifies the Role of PRCs in Aging”: Should read "Given that the above study indicated…"

We have fixed it.

12) Figure 1C and D: it is unclear why red highlighted area extends below diagonal line, if the authors claim that signals in aged animals are higher than in younger? (same goes for some of the other scatter plots…)

The highlighted area is the contour plot, which represents the density function of the scattered points. As shown in Figure 1C that majority (75%) of the points are above the diagonal line, and this shift is further enhanced with age (Figure 1D). While some of the points fall below the diagonal line, our data supported the conclusion that H3K27me3 signals in WT had an age-associated progressive increase.

13) Figure 4C: what is the age of analyzed animals?

That was 30d old animals. We apologize for this. In revised manuscript, we intended to clearly label age of the samples used next to the figure panel, in the main text, and figure legends.

14) Locomotion: usually longer distances than 2cm are recorded (e.g. PMID:25850672 uses 8cm in 20 seconds). Any specific reason for short distance?

We thank the reviewer for raising this point. We noted that Figueroa-Clarevega and Bilder (2015) used 8cm distance to measure both climbing ability and climbing speed (Figueroa-Clarevega and Bilder, 2015). Compared to their settings, our locomotion experiment (2cm) was simplified as we only aimed at discerning whether climbing ability was modulated with age and in mutants. For the same purpose, Liu et al. used 1.5cm that was sufficient to characterize the climbing ability (Liu et al., 2012).